
# Technical note: Dispersion of cooking-generated aerosols from an urban street canyon

Shang Gao[1], Mona Kurppa[2], Chak K. Chan[1], and Keith Ngan[1]

[1]School of Energy and Environment, City University of Hong Kong, Kowloon, Tat Chee Avenue, Hong Kong
[2]Atmospheric Composition Research, Finnish Meteorological Institute, Helsinki, Finland

**Correspondence:** Chak K. Chan (chak.k.chan@cityu.edu.hk); Keith Ngan (keith.ngan@cityu.edu.hk)

**Abstract.** The dispersion of cooking-generated aerosols from an urban street canyon is examined with building-resolving computational fluid dynamics (CFD). Using a comprehensive urban CFD model (PALM) with a sectional aerosol module (SALSA), emissions from deep frying and boiling are considered for near-ground and elevated sources. It is found that, with representative choices of the source flux, the inclusion of aerosol dynamic processes decreases the mean canyon-averaged number concentration by $15-40\%$ for cooking emissions, whereas the effect is significantly weaker for traffic-generated aerosols. The effects of deposition and coagulation are comparable for boiling, but coagulation dominates for deep frying. Deposition is maximised inside the leeward corner vortices, while coagulation increases away from the source. The characteristic timescales are invoked to explain the spatial structure of deposition and coagulation. In particular, the relative difference between number concentrations for simulations with and without coagulation are strongly correlated with the ageing of particles along fluid trajectories or the mean tracer age.

## 1 Introduction

Computational fluid dynamics (CFD) is a well-established tool for studying urban pollutant dispersion (e.g. Rivas et al., 2019). By including an explicit representation of buildings, urban flows can be simulated more accurately than is possible with coarse-resolution mesoscale atmospheric models (Park et al., 2015) or semi-analytical solutions like the classical Gaussian plume model (Melli and Runca, 1979). Most urban CFD studies assume neutral (uniform density) flow and passive scalar dynamics. In recent years, however, the accuracy of urban CFD models has been increased through the inclusion of additional physical processes such as solar heating (Nazarian and Kleissl, 2016) and gas-phase chemistry (Zhong et al., 2015).

One extension that has received relatively little attention is the inclusion of aerosol dynamic processes. The dynamics of urban aerosols differs from that of completely passive, neutrally buoyant particles because their evolution is affected by processes such as condensation, coagulation, deposition and nucleation (Seinfeld and Pandis, 2016); the importance of coagulation and deposition for urban nanoparticles has been noted by Karl et al. (2016). Since particulate matter poses potentially severe risks to human health (Greene and Morris, 2006), improved modelling of urban aerosols is desirable. Urban CFD studies of aerosols have examined ultra-fine particles (Nikolova et al., 2011; Scungio et al., 2013), PM2.5 evolution (Zhang et al., 2011) and aerosol-chemistry coupling (Kim et al., 2012, 2019) arising from vehicular particle sources. Deposition is usually the only



aerosol process included in urban CFD models as it is the most important for traffic emissions within street canyons (Kumar et al., 2011). Nevertheless, additional aerosol processes have been incorporated into global and regional atmospheric models using sectional aerosol modules in which the distribution of particles is represented with a set of discrete size bins (e.g. Gong et al., 2002). The sectional aerosol module SALSA (Kokkola et al., 2008) was coupled to the PALM urban CFD model by Kurppa et al. (2019) (hereafter K19), yielding good agreement with in situ measurements of the aerosol size spectrum.

Although numerical modelling of urban aerosols has focused on emissions from motor vehicles, other sources, such as ships (Ackerman et al., 1995) and factories (Purba and Tekasakul, 2012) also exist. Cooking-generated aerosols from restaurants can have a surprisingly large effect: in situ measurements conducted in a densely urbanised neighbourhood of Hong Kong show that the contribution of cooking emissions to organic aerosols may exceed that of motor vehicles (Lee et al., 2015; Liu et al., 2018). Despite their importance, little is known about the dynamics of cooking-generated aerosols in the outdoor environment;

current understanding relies on in situ measurements and idealised laboratory experiments (Gao et al., 2015). There are strong reasons for expecting the dispersion of traffic-generated and cooking-generated aerosols to differ qualitatively. First, the size distribution of cooking emissions is shifted towards smaller particles as the proportion of particles with a diameter of O(10 nm) or less is much larger (See and Balasubramanian, 2006; Yeung and To, 2008). Hence the relative importance of the aerosol processes may change. Second, the particles are emitted from kitchen exhaust ducts, which may be located near the ground or

far above it. Given that the dispersion of passive scalars is sensitive to the emission location (Huang et al., 2015; Duan et al., 2019), the aerosol dynamics and concentrations may be strongly affected.

    In this paper, the dispersion of cooking-generated aerosols from a street canyon is analysed with large-eddy simulation (LES) and a sectional aerosol module. After reviewing the methodology (Sect. 2), results are presented for different types of emission scenarios (Sect. 3). The effects on the aerosol dynamic processes are highlighted. The results are analysed in terms

of the underlying aerosol timescales, with particular emphasis on the spatial structure of the aerosol processes, in Sect. 4. The sensitivity to factors such as the source strength and background aerosol concentrations is considered in Sect. 5. Limitations of the study are discussed in Sect. 6; conclusions are given in Sect. 7.

## 2 Methodology

For simplicity, the dynamical core of the CFD model and the aerosol module are described separately.

### 2.1 Numerical formulation

#### 2.1.1 PALM

The parallelized large-eddy simulation model (PALM) (Maronga et al., 2015) is an LES model based on the non-hydrostatic, filtered, incompressible Navier-Stokes equations. The 1.5-order Deardorff subgrid-scale (SGS) scheme (Deardorff, 1980) is used to parameterize SGS turbulent fluxes. Fifth-order differencing (Wicker and Skamarock, 2002) is combined with third-

order Runge-Kutta time-stepping (Williamson, 1980). While LES is more computationally expensive than Reynolds-averaged





Navier–Stokes (RANS), the inclusion of transient dynamics allows for nonlinear aerosol processes to be represented more accurately (see Sec. 5.2).

### 2.1.2 SALSA

SALSA includes representations of condensation, coagulation, nucleation and dry deposition (Kokkola et al., 2008). Following K19, only dry deposition, coagulation and condensation are retained. Nucleation, which is computationally expensive to simulate, is not considered in this work. Details on the implementation are given in Appendix A. Briefly, deposition removes particles near surfaces; coagulation coalesces smaller particles into larger ones, decreasing the number concentration but shifting the size distribution; condensation of gases onto particles gases changes the aerosol concentration. As with other sectional models, the parameterisations depend on the particle size. SALSA divides the size distribution into several subranges, each of which is discretised into a specified number of size bins based on the particle diameter, $D_p$. We adopt the same partitioning as K19: in subrange 1, $D_p \in [3\,\text{nm}, 50\,\text{nm}]$; in subrage 2, $D_p$ >50 nm. The particle number in each size bin, $n_i$, is a prognostic variable. The total particle number $N = \sum_i n_i$.

Particles are introduced via injection of a specific component or through condensation of gaseous components. The chemical components include organic carbon (OC), black carbon (BC), sulfuric acid ($H_2SO_4$), nitric acid ($HNO_3$), ammonium ($NH_3$), sea salt, dust and water ($H_2O$). Subrange 1 includes OC, $H_2SO_4$, $HNO_3$ and $NH_3$ only, which are assumed to be internally mixed; subrange 2 includes all the chemical components. Gaseous concentrations of $H_2SO_4$, $HNO_3$, $NH_3$, semi-volatile (NVOCs) and non-volatile organics (SVOCs) also serve as prognostic variables; however, chemical transformations are excluded.

## 2.2 Configuration

### 2.2.1 PALM

A single street canyon of unit aspect ratio, i.e. with dimensional building height $H = 20$ m and width $W = 20$ m, is located at the centre of the computational domain (Fig. 1a). The domain has dimensions $5H$, $2H$ and $5H$ in the streamwise ($x$), spanwise ($y$) and vertical ($z$) directions, respectively; the spanwise extent is somewhat limited but comparable to that of previous studies (e.g. Baik et al., 2007; Duan et al., 2019). The uniform, isotropic grid spacing $\Delta = 1$ m. The timestep, $\Delta t_{\text{PALM}} \sim 0.1$ s.

The boundary conditions follow previous studies. For the velocity, cyclic boundary conditions are applied in the streamwise and spanwise directions, free slip at the top, and no slip at all solid surfaces. For scalar quantities (including the particle number in each size bin), there are cyclic boundary conditions in the spanwise direction and Dirichlet (e.g. $n_i = 0$) in the streamwise direction. The flow is driven by an external pressure gradient, $dp/dx$ = -0.0006 $\text{Pa}\,\text{m}^{-1}$. The simulations are conducted under neutral conditions with the temperature fixed at 300 K.

The model was spun-up for 1000 s in order to attain a statistically steady flow (as determined from the mean streamwise velocity within the canyon). Subsequently particle emission commenced and the model was run for another 5000 s. Data collected during the last 3000 s (with an output interval of 10 s) are analyzed in this study. During the 3000 s sampling period,





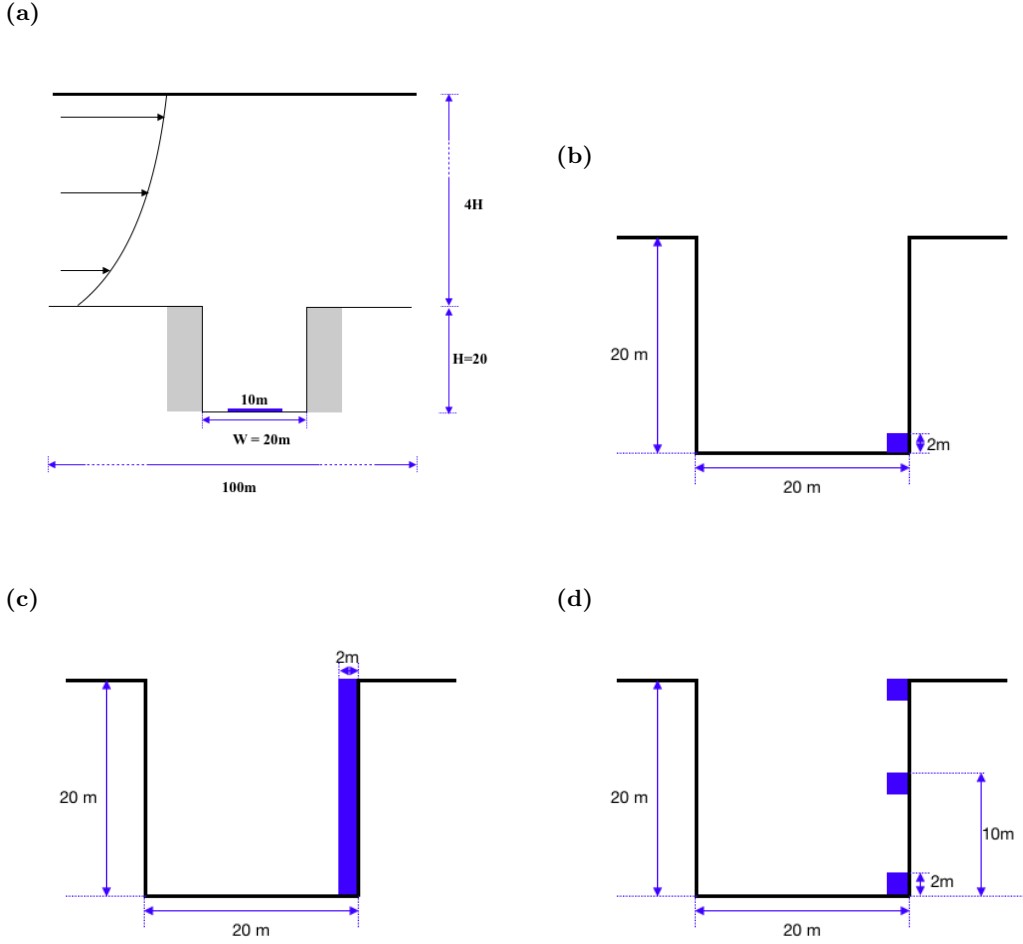

**Figure 1.** Schematic representations in the $x - z$ plane (at the midplane $y/W = 1$) of the computational domain and pollutant sources: **(a)** ground-level traffic emissions; **(b)** near-ground cooking emissions; **(c)** column cooking emissions; **(d)** isolated cooking emissions. The emission scenarios are defined in Table 1. For clarity, the streamwise position of the cooking sources has beeen shifted.

the number concentration also reaches statistical distribution: the ratio of the standard deviation to the mean is 2.7% and 5.9% for runs NG-D and NG-B (Table 1), respectively. The time average is denoted by the overbar. The spanwise average is denoted by $\langle \cdot \rangle$ and the canyon average by $\langle \rangle_C$.

Figure 2 shows mean streamlines and wind speed in the $x-z$ plane. The picture is consistent with the many numerical studies of unit-aspect-ratio street-canyon flow. In particular, there is a large central vortex and smaller canyon vortices. Although the corner vortices are not defined as clearly as in higher resolution simulations (e.g. Duan et al., 2019), recirculations clearly exist within the bottom corners. Ramifications of the streamline topology are considered in Sect. 3.





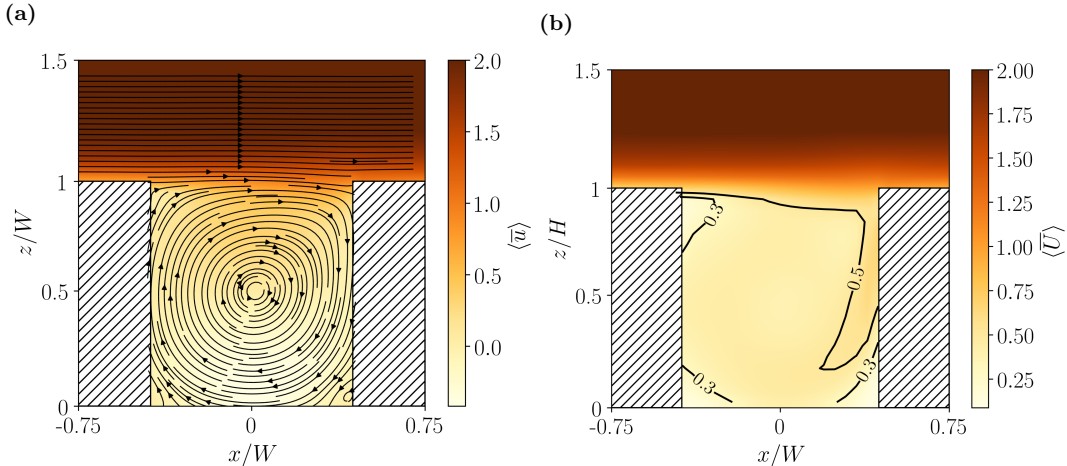

**Figure 2.** Spatial structure of the mean flow in the $x - z$ plane: **(a)** streamwise velocity component, $u$ ($\mathrm{m\,s^{-1}}$); **(b)** wind speed, $U$ ($\mathrm{m\,s^{-1}}$).

### 2.2.2 SALSA

Version 4481 of SALSA is used. Following K19, there are five size bins for each of the two subranges. As the current study is concerned with idealised emission scenarios, the background number concentration for the $i$th size bin, $n_{b,i} = 0$; the sensitivity to $n_{b,i}$ is examined in Sect. 5.1. As in K19, the SALSA timestep $\Delta t_{\mathrm{SALSA}} = 1$ s in all cases. The three aerosol processes in SALSA (coagulation, condensation and deposition) may be activated or deactivated independently of each other.

Pollutants are emitted from uniform area sources. Two basic source types are considered: (i) ground-level traffic; and (ii) cooking emissions from one side of a street canyon. In the former case, a uniform area source is located at the bottom of the canyon. In the latter case, the sources, which cover a portion of the walls facing the street, are located at the roadside (Fig. 1b); between the ground and the roof level (Fig. 1c); and at the bottom, middle or top floors (Fig. 1d). No attempt is made to represent exhaust ducts. These sources represent roadside and elevated (aligned in a column or else isolated) kitchens. In addition, there are two possible cooking modes, namely deep frying and boiling: both are considered as their emission spectra are rather different (Fig. 3). The emission scenarios described above are summarised in Table 1.

For each source type, the emission spectra and source flux ($\mathrm{m^{-2}\,s^{-1}}$) must be specified. Values that are broadly representative of large cities are chosen for the latter. The sensitivity to the net source flux is quantified in Sect. 5.2.

1. *Ground-level traffic (Case TR).* The emission factor for the number of particles emitted by a vehicle per unit distance travelled is $3.0 \times 10^{14}\,\mathrm{km^{-1}\,veh^{-1}}$ (Fujitani et al., 2020). A traffic volume of 1000 vehicles per hour, which corresponds to moderately heavy traffic within a city centre is assumed. The total particle flux ($\mathrm{s^{-1}}$), $T_p$, is obtained from the emission factor and the length of the street, i.e. $T_p = \epsilon L$, whence the source flux $Q = T_p/A$, where $A$ is the area covered by the traffic.



**Table 1.** Emission scenarios for the area sources illustrated in Fig. 1. The dimensions of the area sources are expressed in terms of the canyon dimensions $(W, L, H)$ when the source extends along the full extent of the canyon in a given direction, and $(w, l, h)$ otherwise. For the isolated cooking emissions, $z_0/H \in \{0.05, 0.50, 0.95\}$.

| Case | Type | Location | Source dimensions |
|------|------|----------|-------------------|
| TR | Traffic | Ground level (centred at $x/W = 0$) | $w = 10$ m, $L$ |
| NG-D | Deep frying | Near-ground | $L$, $h = 2$ m |
| NG-B | Boiling | Near-ground | $L$, $h = 2$ m |
| CO-D | Deep frying | Column (centred at $y/W = 1$) | $l = 2$ m, $H$ |
| CO-B | Boiling | Column (centred at $y/W = 1$) | $l = 2$ m, $H$ |
| I-D-$z_0$ | Deep frying | Isolated (centred at $y/W = 1, z/H = z_0$) | $l = 2$ m, $h = 2$ m |
| I-B-$z_0$ | Boiling | Isolated (centred at $y/W = 1, z/H = z_0$) | $l = 2$ m, $h = 2$ m |

2. *Cooking emissions.* The emission factors for the number of particles emitted per unit time by a kitchen of unit volume are $3.75 \times 10^{10}\,\mathrm{m^{-3}\,s^{-1}}$ and $4.31 \times 10^{9}\,\mathrm{m^{-3}\,s^{-1}}$, for deep frying and boiling, respectively. These values are derived from data for reference kitchens with a volume of $\sim 20$ m$^3$; it is assumed that no particles are trapped indoors. The total particle flux (s$^{-1}$) is obtained from the emission factor and the volume of the kitchens, i.e. $T_p = n\epsilon V$, where $V$ is the volume and $n$ is the number of kitchens. It is assumed for simplicity that particles are emitted uniformly over the external face of the restaurant. The source flux follows from normalisation by $A$, the area of the kitchen face parallel to the canyon axis. The indoor-outdoor exchange is entirely one-way: particles escape from the kitchens to the outdoor environment but no particles travel in the opposite direction.

The source specification depends on the emission scenario (Table 1):

- *Near-ground emissions (Case NG).* It is assumed that each roadside kitchen has dimensions $4\,\mathrm{m} \times 2\,\mathrm{m} \times 2\,\mathrm{m}$, with the longest side being parallel to the street. Hence the total particle flux equals the combined emissions from 10 restaurants.

- *Column emissions (Case CO).* There are a total of five elevated kitchens between the bottom and top floors. The kitchens may be taken to be domestic rather than commercial and of smaller dimensions, namely $2\,\mathrm{m} \times 2\,\mathrm{m} \times 4\,\mathrm{m}$. For simplicity, the combined emissions from the separate kitchens are represented by a continuous column source.

- *Isolated emissions (Case I).* Only a single elevated kitchen is considered. It has dimensions $2\,\mathrm{m} \times 2\,\mathrm{m} \times 4\,\mathrm{m}$.

Emission spectra are obtained by scaling the reference spectra (Fig. 3); the contribution of a specific size bin is determined by $T_p$ and the emission spectrum. No attempt is made to represent the kitchen ventilation system; hence all indoor aerosol processes are excluded. The scaling factor for a size bin is given by the ratio of the integral of the emission spectrum over the size bin to the integral over the entire spectrum. Note that, for a kitchen of identical size, deep frying generates nearly 10 times as many particles as does boiling. In addition, more small particles are generated for deep





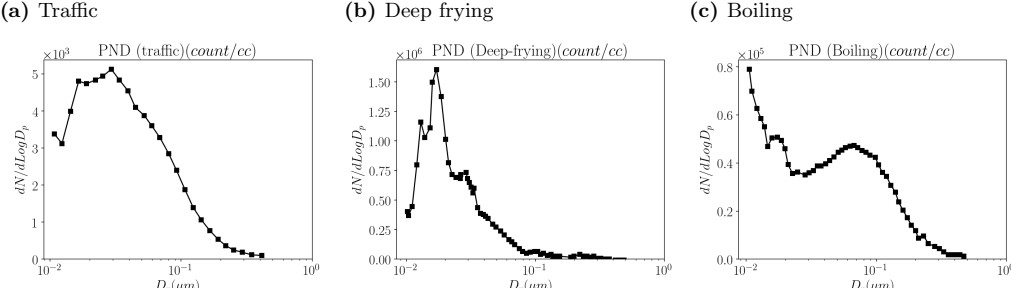

**Figure 3.** Reference emission spectra: **(a)** traffic (Janhäll et al., 2004); **(b)** deep frying (See and Balasubramanian, 2006); **(c)** boiling (See and Balasubramanian, 2006). The reference spectra are scaled in order to obtain emission spectra for the scenarios of Table 1.

frying: the proportion of nanoparticles in the 1 nm to 100 nm range increases from 65% to 90% and the mean diameter decreases from 57.4 nm to 26.5 nm (See and Balasubramanian, 2006; Yeung and To, 2008). However, the boiling spectrum peaks at a smaller diameter.

As described above, several assumptions are required to specify the particle emissions from cooking; their validity is discussed in Sec. 6. The gaseous emissions are specified ine Appendix B: in particular, the chemical compositions (Table B-1) and emission factors (Table B-2) are listed.

### 2.3 Validation

Mean velocity statistics are validated against the wind-tunnel data of Brown et al. (2001) in Fig. 4. The vertical profiles of the mean streamwise velocity $u$ and mean wind speed $U = \sqrt{u^2 + v^2 + w^2}$ show good agreement. This is confirmed with standard validation metrics (Appendix C): the normalised mean square error $NMSE \sim 0.01 - 0.04$, fractional bias $FB \sim 0.02$ and correlation coefficient $R \sim 0.99$; for a perfect validation, $NMSE = FB = 0$ and $R = 1$. The agreement is comparable to previous numerical studies (Cui et al., 2004; Duan et al., 2019).

Although passive scalar statistics have been successfully validated (Appendix C), this is of secondary relevance for aerosol modelling. Following K19, the coupled PALM-SALSA model is validated against evening measurements of the aerosol number concentration within a real street canyon in Cambridge, UK (Kumar et al., 2008). For simplicity, the computational domain of dimensions $167 \text{ m} \times 60 \text{ m} \times 60 \text{ m}$ contains a single street canyon of $167 \text{ m} \times 12 \text{ m} \times 12 \text{ m}$ and no other buildings. The boundary conditions follow Sect. 2.2.1. Using the traffic data in K19, emissions from the street canyon only are considered. Vertical profiles of the aerosol number concentration from the current LES show improved agreement for $z/H \lesssim 0.6$ compared to K19. Exact agreement cannot be expected because the computational domain is much smaller in the present validation (and emissions from outside the street canyon neglected). Nonetheless, we conclude that the present configuration is capable of reproducing the aerosol distribution in the real urban environment.



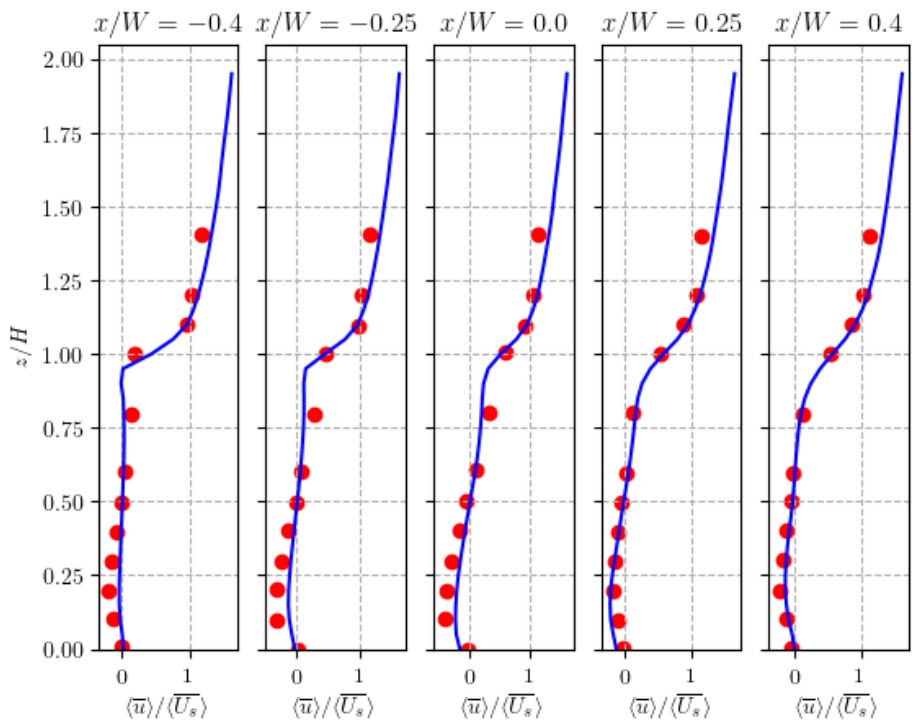

**Figure 4.** Vertical profiles of the normalised mean streamwise velocity, $\langle \overline{u} \rangle / \langle \overline{U_s} \rangle$, for the present LES and wind-tunnel measurements (Brown et al., 2001). $\langle \overline{U_s} \rangle$ is the average streamwise velocity within the shear layer ($1 \leq z/H \leq 1.5$). The LES results are plotted with blue lines, the wind-tunnel data with red circles.

## 3 Results

### 3.1 Traffic and near-ground cooking emissions

To highlight the influence of the emission spectrum, we begin by comparing the aerosol number concentration fields generated by traffic and roadside restaurants, i.e. emission scenarios TR, NG-D and NG-B (Table 1). Fig. 6 plots spanwise averages of

160 the dimensionless mean concentrations,

$$\overline{N_*} = \frac{\overline{N} U_{\text{ref}} H L}{T_p}, \tag{1}$$

where $\bar{N}$ is the time-averaged total number concentration ($\text{m}^{-3}$) and $U_{\text{ref}}$ ($\text{m s}^{-1}$) is the streamwise velocity at $2.5H$. In the absence of aerosol dynamic processes (NOAD, left panels), the number concentration is essentially a passive scalar. For traffic emissions from the ground-level source (Fig. 6a), there are elevated concentrations within and around the vortex at the bottom

leeward corner, as has been observed in many studies (for the related case of a line source see Pavageau and Schatzmann, 1999). For roadside cooking emissions from the windward side (Figs. 6b,c), pollutants recirculate around the corner before they can disperse through the rest of the canyon. Similar results for a pair of line sources were obtained by Huang et al. (2015);





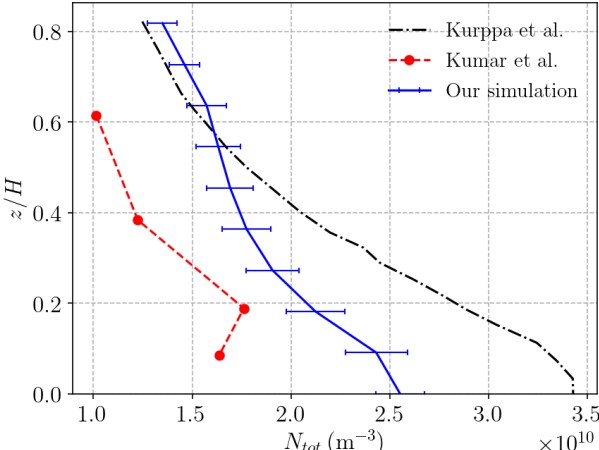

**Figure 5.** Vertical profiles of the aerosol number concentration within a street canyon in Cambridge, UK. The original in situ evening measurements (Kumar et al., 2008) are plotted in red. LES simulations using PALM-SALSA from K19 and the current study are plotted in black and blue, respectively. Error bars (standard deviations) for the current LES are plotted with horizontal lines.

the fluid-dynamical processes governing escape from the corner vortex are discussed by Duan et al. (2019), who analysed the initial-value problem rather than the forced one. Deep frying (NG-D) and boiling (NG-B) yield identical concentrations in the absence of aerosol dynamic processes. In all cases, the concentration field reflects the combined influence of the mean flow (streamline geometry) and source location. Canyon-averaged number concentrations are summarised in Table 2.

**Table 2.** Canyon-averaged dimensionless concentrations, $\langle \bar{N} \rangle_C$, for the near-ground (Fig. 6) and column (Fig. 7) cooking emissions. The errors correspond to the spatial standard deviation; the relative change in the mean concentrations is also listed. Both the mean and standard deviation are time-averaged.

|  | Near-ground | | | Column | | |
|---|---|---|---|---|---|---|
|  | NOAD | AD | difference | NOAD | AD | difference |
| Boiling | $144.4 \pm 5.4$ | $122.9 \pm 3.5$ | -15% | $193.1 \pm 7.7$ | $178.4 \pm 5.8$ | -8% |
| Deep frying | $144.4 \pm 5.4$ | $87.2 \pm 1.4$ | -40% | $193.1 \pm 7.7$ | $154.5 \pm 2.5$ | -20% |
| Traffic | $69.1 \pm 2.4$ | $67.7 \pm 2.1$ | -2% | - | - |  |

The effect of aerosol dynamic processes (AD) is illustrated by the right panels. For traffic emissions, the spatial structure of the number concentration field is almost identical (Fig. 6b) while the canyon-averaged and pedestrian-level concentrations decrease by around 2%. In their study of a neighbourhood in Cambridge, UK, K19 found that aerosol processes cause the number concentration to decrease by $\sim 10\%$. One possible explanation for this discrepancy is that the emission spectra differ: the mean particle size is larger in the current study, i.e. 47.9 nm rather than 32.7 nm. This is significant because smaller





**Figure 6.** Normalised mean aerosol number concentration, $\langle\overline{N_*}\rangle$, for different emission scenarios: (a,b) traffic (Case TR); (c,d) near-ground, deep frying (Case NG-D); (e,f) near-ground, boiling (Case NG-B). Results without **(NOAD)** and with **(AD)** aerosol dynamic processes are shown at the left and right, respectively. The approximate position of the roadside restaurants is indicated by the white lines (which are shifted for clarity).





particles may have a larger deposition velocity, leading to enhanced deposition (see Sect. 3.3.2 for further discussion). For cooking emissions, the spatial structure of the concentration fields and the mean values change. With deep frying (Fig. 6d), the highest concentrations (shown in red) are confined to a smaller region in the windward corner and the isolines are strongly

perturbed. With boiling (Fig. 6f), the highest concentrations are confined to a larger region and the isolines are deflected, though not as dramatically as for deep frying. The canyon-averaged concentrations decrease by 15% for boiling and 40% for deep frying. Since cooking, whether through boiling or deep frying, generates more small particles than does traffic (Fig. 3), the coagulation and dry deposition rates should be higher.

Of course the results will differ with other assumptions about the kitchen dimensions (Sect. 2.2.2). The sensitivity to the

source flux is examined in Sect. 5.2.

For reference, mass concentration fields are shown in Appendix D. In general, PM2.5 mass concentrations (i.e. for particles smaller than 2.5 $\mu m$ in diameter) are higher for cooking emissions, especially for deep frying. Indeed the maximum concentration for NG-D reaches $200 \mu g$ m$^{-3}$ near the source region (Fig. D-1); possible reasons for these high values are given in Sect. 6. In agreement with the measurement campaign of Lee et al. (2015), the local contribution of cooking emissions exceeds

that of traffic. However, the mass concentration shows much less sensitivity to the inclusion of aerosol processes.



## 3.2 Elevated kitchens

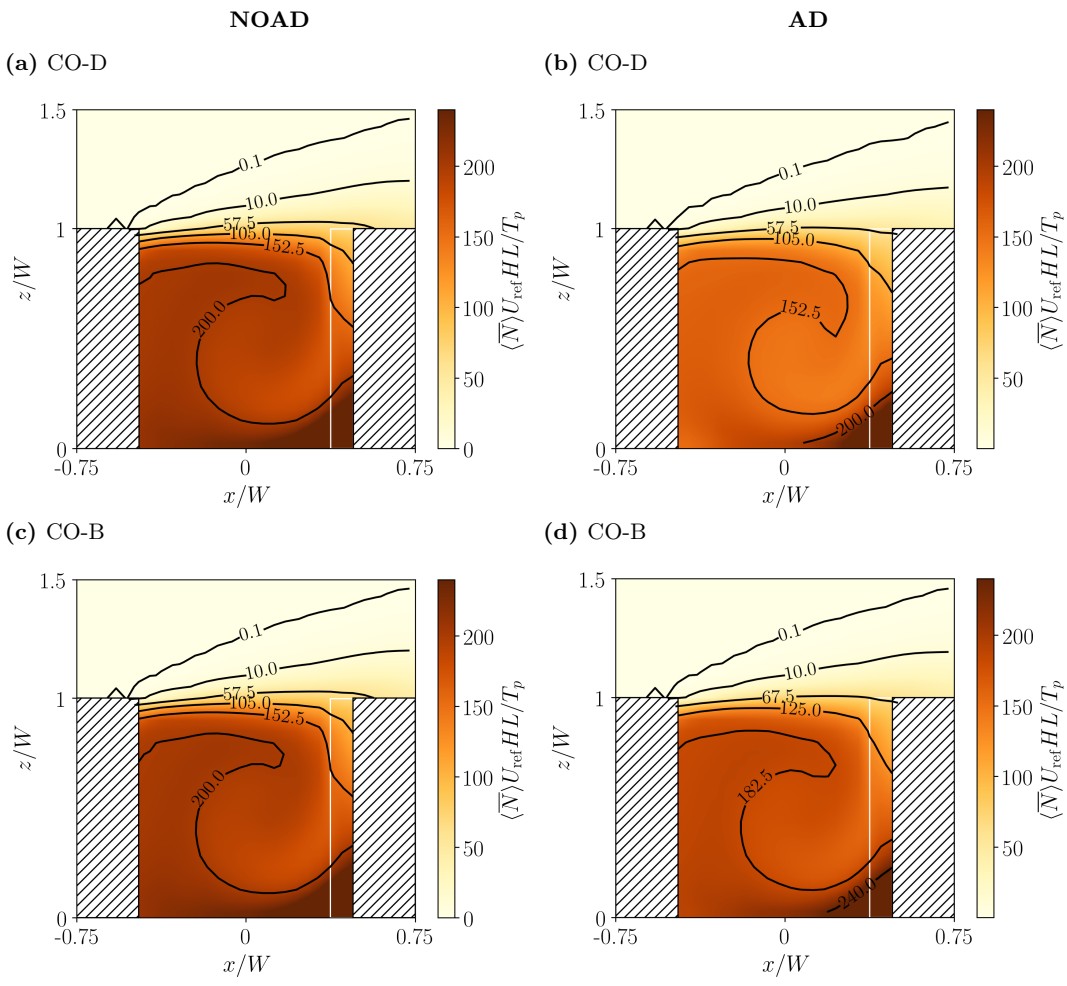

**Figure 7.** As in Fig. 6, but for column cooking emissions: (a,b) deep frying; (c,d) boiling.

Kitchens may not be located at the roadside. This section considers continuous emissions along a column extending from the bottom to top of a building, as well as isolated kitchens on different levels.

The qualitative effect of column cooking emissions resembles the near-ground emissions. For both deep frying and boiling, the familiar pattern of very high concentrations in the bottom leeward corner and lower concentrations aloft is maintained (Fig. 7); however, the concentrations increase in the interior. Averaged over the canyon, the spatial mean and standard deviation increase for column emissions (Table 2), but the sensitivity to aerosol processes weakens: the difference between NOAD and AD is 8% and 20% for boiling and deep frying, respectively. If the number concentration depended linearly on particle emissions, the nondimensionalisation would remove the dependence on the total particle flux, $T_p$; nonlinear effects are discussed in Sect. 5.1.





The influence of the source height is stronger for the isolated kitchens. Although trapping of particles within the vortex at the bottom leeward corner is less evident (Figs. 8b,c), emission from elevated sources actually increases the canyon-averaged concentrations (Table 3). This is consistent with elevated concentrations around the central canyon vortex (Duan et al., 2019).

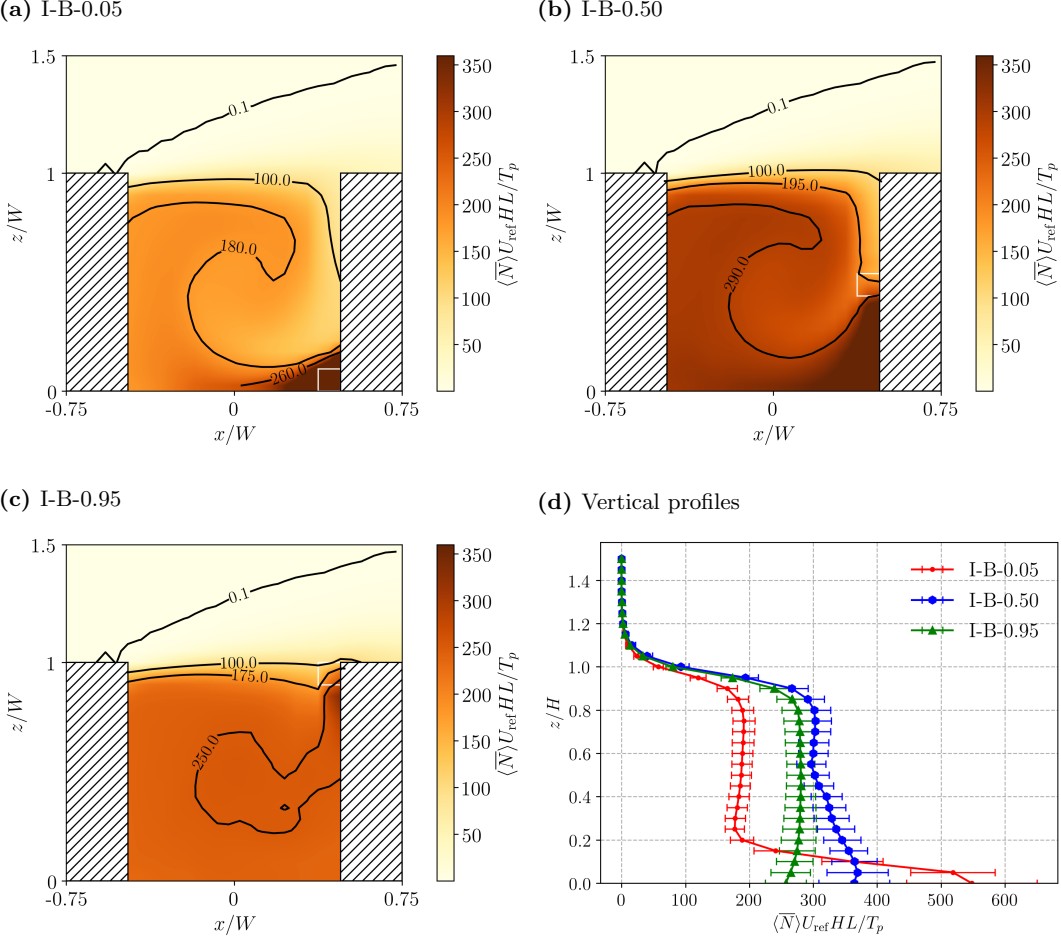

**(a)** I-B-0.05      **(b)** I-B-0.50

**(c)** I-B-0.95      **(d)** Vertical profiles

**Figure 8.** As in Fig. 6, but for isolated kitchens.

**Table 3.** As in Table 2, but for deep-frying emissions from isolated kitchens.

|  | NOAD | AD | difference |
|---|---|---|---|
| I-D-0.05 | $219.3 \pm 7.2$ | $181.3 \pm 3.0$ | -17% |
| I-D-0.50 | $289.9 \pm 10.8$ | $264.4 \pm 4.7$ | -9% |
| I-D-0.95 | $242.5 \pm 13.8$ | $232.4 \pm 4.3$ | -4% |



### 3.3 Comparison of aerosol dynamic processes

The effect of the individual aerosol processes is assessed by analysing separate SALSA configurations in which condensation, coagulation, deposition, or all three processes acting simultaneously, are enabled. As in K19, the relative difference,

$$RD_i = \frac{\langle \overline{N_{NOAD}} \rangle - \langle \overline{N_i} \rangle}{\langle \overline{N_{NOAD}} \rangle}, \qquad (2)$$

is defined from the mean concentrations with aerosol processes, $\langle \overline{N_i} \rangle$, and without them, $\langle \overline{N_{NOAD}} \rangle$. In the former case, $i \in \{\mathrm{COND, COAG, DEPO, ALL}\}$ labels the different SALSA configurations. The subscript is dropped when there is no risk of confusion. For brevity, not all of the emission scenarios listed in Table 1 are analysed here. Results for boiling may be found in Appendix E.

### 3.3.1 Traffic emissions

Figure 9 compares the effects of the different aerosol processes for traffic emissions. Vertical profiles of the mean number concentration have nearly identical shapes for the different configurations (Fig. 9a). The lowest concentrations are obtained for deposition only. On account of nonlinearity, the effects do not add linearly. By contrast with K19, who considered a domain with uneven building heights, the relative difference shows minimal variation with height (Fig. 9b). Condensation has a negligible effect on the aerosol number concentration because it primarily serves to increase the volume of particles (Seinfeld and Pandis, 2016). The effects of deposition and coagulation are approximately constant away from the bottom boundary. For $z/H > 0.2$, $RD \sim 4.5\%$ for deposition and $\sim 0.4\%$ for coagulation. The estimate of $RD_{DEPO}$ is low compared to K19, who obtained $RD_{DEPO} \sim 15\%$ using a different emission spectrum, non-zero background concentration and realistic urban topography. Nonetheless, deposition remains the most important process for traffic emissions, in agreement with the timescale analysis of Ketzel and Berkowicz (2004).

We now consider the spatial structure of the two most important processes. For deposition (Fig. 9c), RD is maximised in the bottom corners. Values are lower away from the corners, especially near the centreline, $x/W = 0$. For coagulation (Fig. 9d), by contrast, the pattern is rather different: RD is maximised within the bottom windward vortex and the central canyon vortex. Roughly speaking, the effects of deposition are fairly small outside the bottom corner vortices, while those of coagulation tend to increase away from the source centred at $x/W = 0$. The increase follows the orientation of the mean circulation (cf. Fig. 2a).

### 3.3.2 Near-ground cooking emissions

The effects of the different aerosol processes for near-ground deep drying are compared in Fig. 10. Compared with traffic emissions, the effects of coagulation are much more important: the vertical profiles for COAG and ALL nearly coincide (Fig. 10a) and $RD_{COAG}$ reaches a maximum of around 40% (Fig. 10b), which is around 400 times higher than that for traffic emissions. $RD_{DEPO}$ also increases significantly. There are several reasons for these differences. First, coagulation occurs more efficiently for particles with $D_p < 50$ nm (Kokkola et al., 2008). Such particles are favoured by the emission spectrum (93% of the particles generated by deep frying fall in this category, but just 58% for traffic; Fig. 3). Second, the efficiency





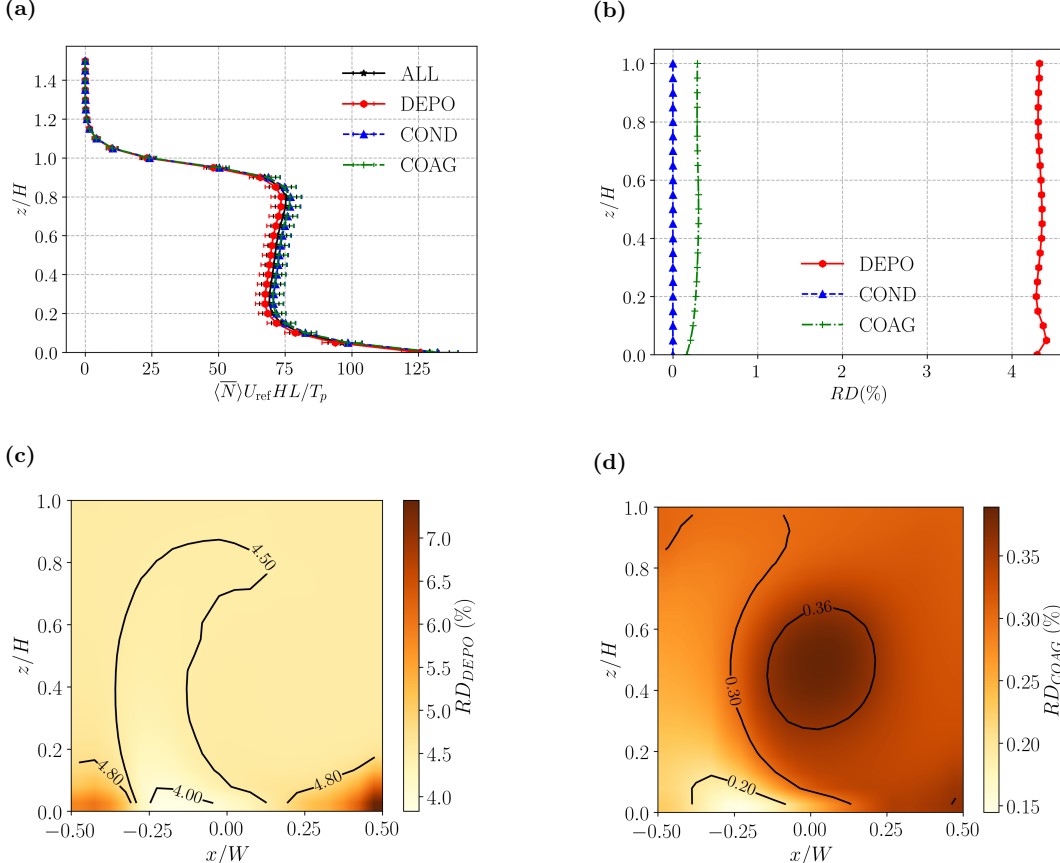

**Figure 9.** Effect of different aerosol processes for Case TR. Mean vertical profiles of **(a)** mean aerosol number concentration; **(b)** relative difference with respect to a simulation without aerosol dynamic processes (NOAD). Spanwise-averaged relative difference fields for **(c)** deposition; **(d)** coagulation. The lines correspond to different SALSA configurations (DEPO – deposition only; COND – condensation only; COAG – coagulation only; ALL – deposition, condensation and coagulation).

of Brownian diffusion increases for smaller particles, leading to higher deposition velocities and enhanced deposition (Zhang et al., 2001; Kurppa et al., 2019). Above $z/H \sim 0.2$, which lies above the corner vortices in Fig. 2, the RD profiles are nearly independent of height.

The spatial structure of the RD fields also changes. For deposition (Fig. 10c), RD is maximised in the bottom leeward corner, while the lowest values are found in the bottom windward corner. (RD gives an exaggerated impression of the absolute difference between the corners as the total number concentration for NOAD is approximately 50% higher in the windward
corner.) For coagulation (Fig. 10d), the lowest values are no longer found around the source centred at $x/W = 0$, as is the case for traffic (Fig. 9d), but rather around the roadside kitchens on the windward wall. Once again, there is indication that the





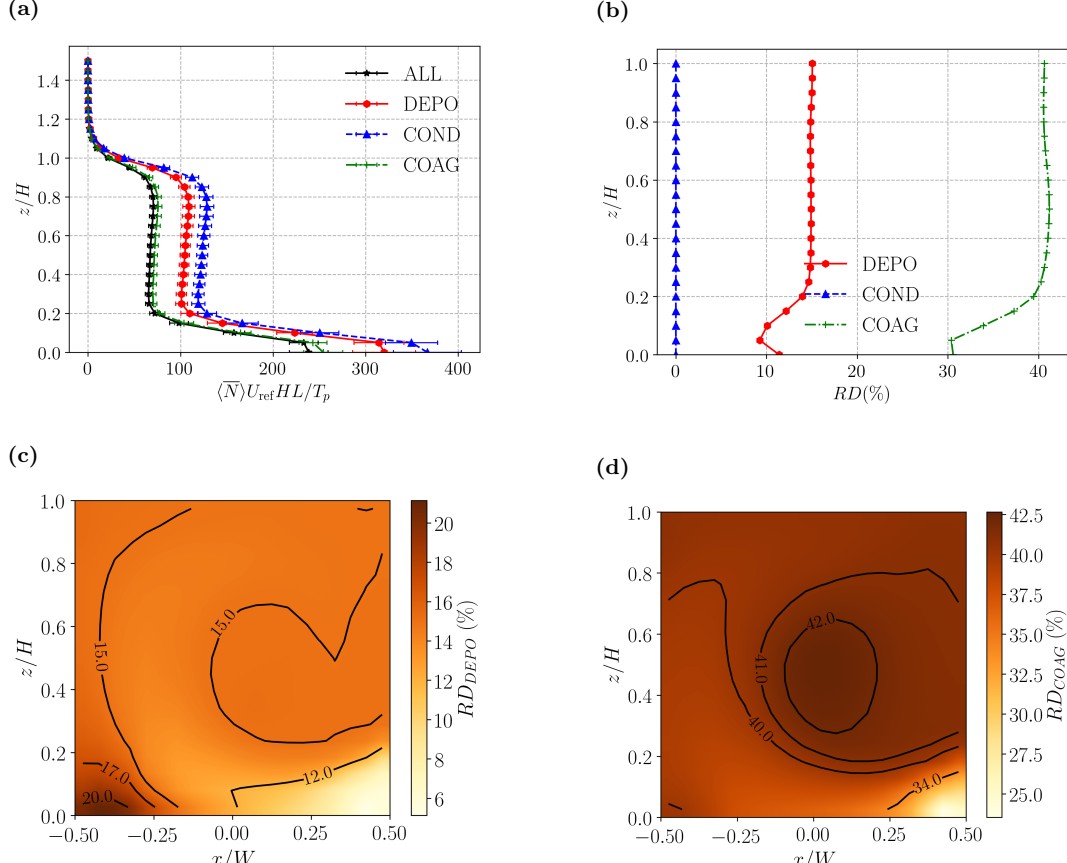

**Figure 10.** As in Fig. 9, but for Case NG-D.

relative importance of coagulation increases away from the source following the sense of the mean circulation. The largest RD values are found near the centre.

Qualitatively similar results are obtained for Case NG-B (Fig. E-1). While number concentrations and RD values are lower (cf. Table 2), the structure of the RD fields is largely unchanged from Fig. 10.

### 3.3.3   Column cooking emissions

The effects of the different aerosol processes for column deep frying are compared in Fig. 11. Although the emission spectrum is unchanged from Case NG-D (Fig. 10), both $RD_{DEPO}$ and especially $RD_{COAG}$ decrease (Fig. 11b). This change cannot
be directly attributed to the increase in the canyon-averaged number concentration (Table 2). Everything else being the same, the deposition rate should scale linearly with $\bar{N}$ (Sect. A.1), while the coagulation should scale quadratically (Sect. A.2). Given the nondimensionalisation and the increase in $\bar{N}$, one would naively expect $RD_{DEPO}$ to be approximately unchanged and $RD_{COAG}$ to increase. Evidently the change in the source configuration, rather than the associated increase in number





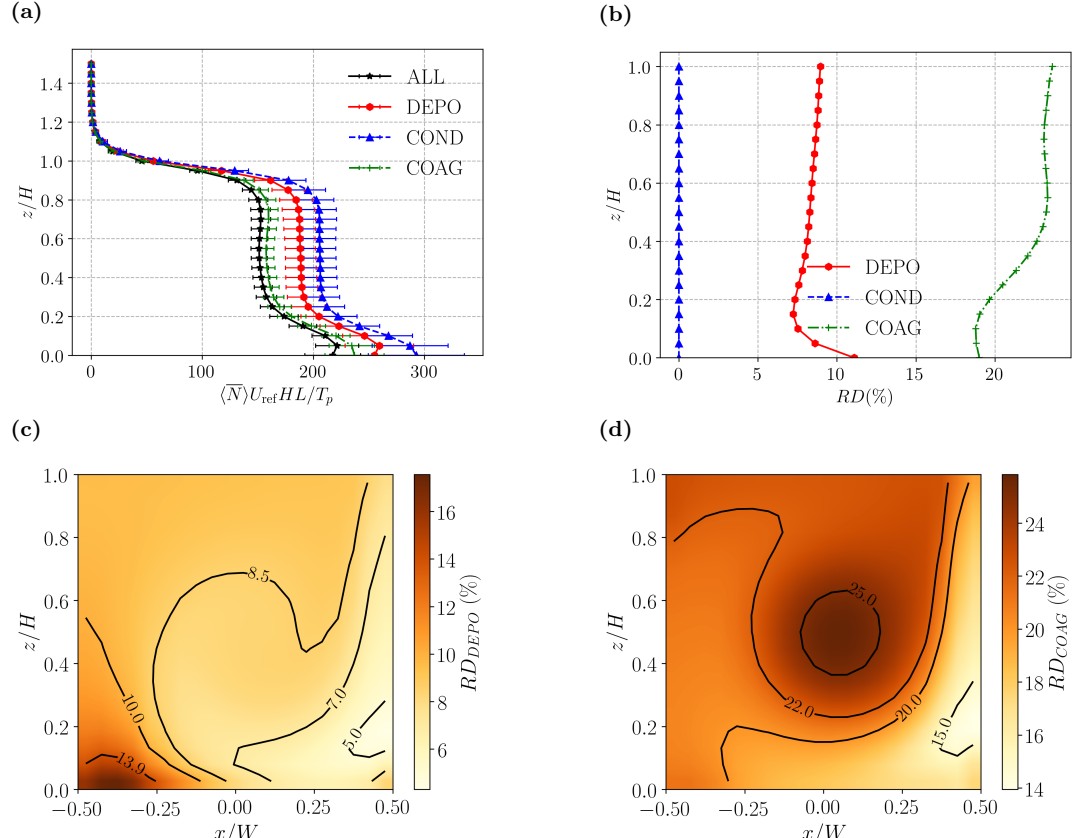

**Figure 11.** As in Fig. 9, but for Case CO-D.

concentration, is of primary importance. The spatial distribution provides partial insight into this. Analogously to NG-D,

$RD_{DEPO}$ and $RD_{COAG}$ have relatively low values along the windward wall (Figs. 11c,d); however, the column source covers a larger area and a plume-like structure develops away from it. Results for CO-B are qualitatively similar (Fig. E-2).

### 3.4 Aerosol number distributions

Aerosol size distributions for ground-level traffic and near-ground cooking are compared in Fig. 12 for different emission scenarios and SALSA configurations. The statistics are evaluated over the entire canyon. Deviations with respect to NOAD

reflect the influence of aerosol processes. For Case TR (Fig. 12a), deposition decreases the number of particles in each size bin for $D_p < 50$ nm, but has minimal effect for larger particles. The shape is not exactly preserved because deposition is size-dependent (cf. Sect. A.1). The size spectra for COAG and NOAD are almost identical. The effect of coagulation on the size spectra is more obvious for cooking emissions. For Case NG-D (Fig. 12b), the COAG and ALL spectra are nearly identical; the DEPO concentrations are higher at small scales, $D_p < 40$ nm. For Case NG-B (Fig. 12c), the pattern is similar. The effects of





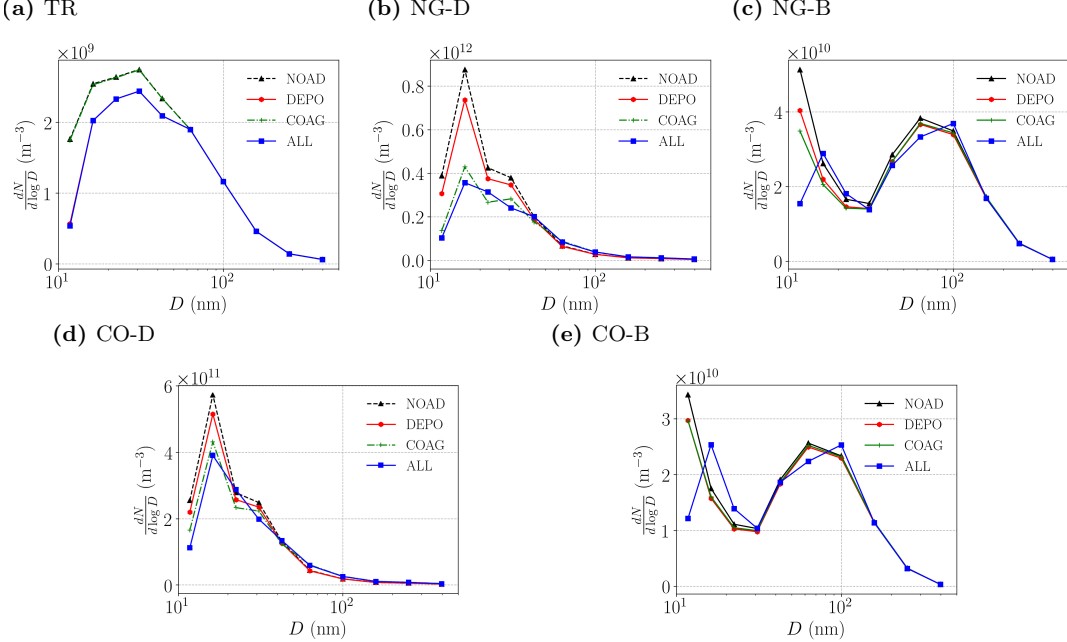

**Figure 12.** Aerosol number size distributions for different emission scenarios: (a) traffic; (b) near-ground deep frying; (c) near-ground boiling; (d) column deep frying; (e) column boiling. Each line corresponds to a different set of aerosol processes (see Fig. 9 for definitions).

COAG are largest for small particles on account of the emission spectra and the increase in coagulation efficiency for smaller particles.

Similar results are obtained for column emissions. For Case CO-D (Fig. 12d), differences with respect to NOAD are smaller, in agreement with the canyon-averaged concentrations, but compared to NG-D, the range over which coagulation is strongest narrows to $D_p \leq 30$ nm. For Case CO-B (Fig. 12e), coagulation leads to decreased concentrations for small particles ($D_p < 20$ nm), as with NG-B.

## 4 Analysis of the aerosol processes

### 4.1 Characteristic timescales

The differences between the aerosol processes can be understood by referring to the characteristic timescales (Fig. 13). For concreteness, we focus on NG-D. The deposition timescale is derived from the deposition velocity, eq. (A1a), which is $O(10^{-2})\,\mathrm{m\,s^{-1}}$ Hence the deposition timescale for the smallest resolved scale, $\tau_{depo} = \Delta/v_d \sim 100$ s, while for the corner vortices, $\tau_{depo} = O(500)$ s. Following Ketzel and Berkowicz (2004), the coagulation timescale,

$$\tau_{coag} = N / \frac{\partial N}{\partial t} \big|_{coag}, \tag{3}$$





**(a)**

**(b)**

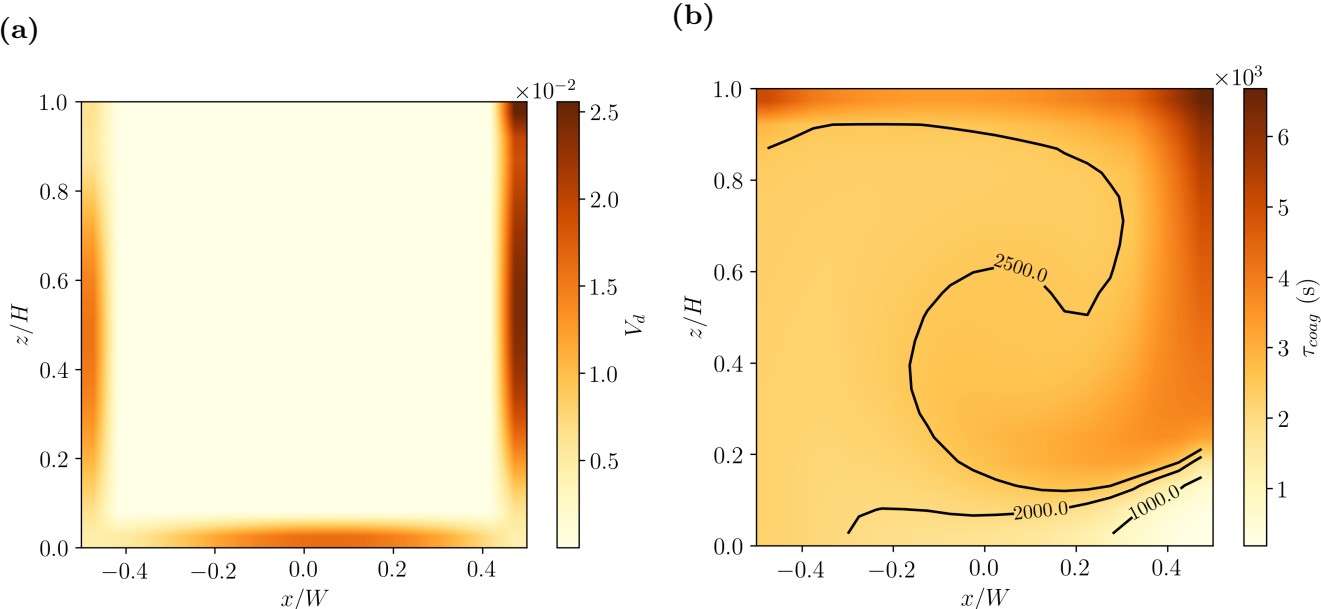

**Figure 13.** Comparison of characteristic timescales for NG-D. (a) $v_d$ (or $\Delta/\tau_d$); (b) $\tau_{coag}$.

may be diagnosed from the evolution equation,

$$\frac{\partial N}{\partial t}\Big|_{coag} = \frac{1}{2}\sum_{j=1}^{k-1}\beta_{k-j,j}n_{k-j}n_j - \sum_{j=1}^{\infty}\beta_{k,j}n_k n_j \tag{4}$$

where $\beta_{ij}$ is the coagulation kernel. Although this neglects the actual time discretisation (cf. Sect. A.2), the error should be negligible so long as $\Delta t \ll \tau_{coag}$. Using the same $\beta_{ij}$ as SALSA and $\bar{n}_i$, the time-averaged number concentrations, $\tau_{coag} \sim 1000-5000$ s for NG-D (Fig. 13b). However, this estimate is somewhat misleading because $\tau_{coag}$ is not constant. From integration of eq. (4), the total number concentration actually decreases from its initial value (i.e. the time-averaged $n_i$) by a factor of $e$ after $\sim 1000$ s (not shown). This suggests that coagulation occurs on a timescale that is roughly comparable to that

of the mean circulation, i.e. $T_c = 2(H/W + U/W) = 382$ s, where $U$ and $W$ are characteristic streamwise and vertical speeds.

Given these estimates for $\tau_{depo}$ and $\tau_{coag}$, several predictions about deposition and coagulation can be made. Since $\tau_{depo}$ is relatively long, deposition will preferentially occur where particles can remain close to solid surfaces for an extended duration; the corner vortices are good candidates because particles may be brought near the walls as they recirculate. This is consistent with $RD_{DEPO}$ (Fig. 10b). Away from the walls, $\tau_{depo}$ is not uniform in the interior because particles are mixed with ambient

fluid. Since $\tau_{coag} \gtrsim T_c$, coagulation is not independent of the mean canyon motion. This means that coagulation proceeds while particles are being advected and mixed. One therefore expects the structure of $RD_{COAG}$ to resemble that of the steady passive scalar field (compare Figs. 6c and 9d). More precisely, $RD_{COAG}$ should be correlated with the age of fluid parcels or the time elapsed between the release of a particle at the source and its arrival at a receptor point. Physically, $RD_{COAG}$ increases away from the source (i.e. as the age increases) because there is more time for coagulation to occur.





Similar considerations apply to other emission scenarios. For NG-B, $v_d$ is comparable but $\tau_{coag} \sim 10000 - 50000$ s (with an $e$-folding time of 7000 s). The implication is that the spatial structure of $RD_{DEPO}$ and $RD_{COAG}$ should be largely unchanged. Deposition is strongly affected by the emission spectrum (cf. Fig. 12), but it should continue to be maximised in the same places for identical flow and source location. Coagulation is much slower compared to NG-D and $RD_{COAG}$ reduced, but the basic pattern should be unchanged so long as coagulation is a relatively slow process governed by the age along fluid trajectories.

These predictions are consistent with Fig. E-1. For CO-B, $v_d$ and $\tau_{coag}$ change little from NG-B (by $3.1\%$ and $4.0\%$). As before, one expects deposition to increase within the corner vortices, though the effect on $RD_{DEPO}$ should be less noticeable within the windward vortex, where concentrations are higher around the source (Fig. 7c). Coagulation should decrease for CO-B on account of the shorter mean ages associated with elevated source locations (see Sect. 4.2).

## 4.2    Mean tracer age

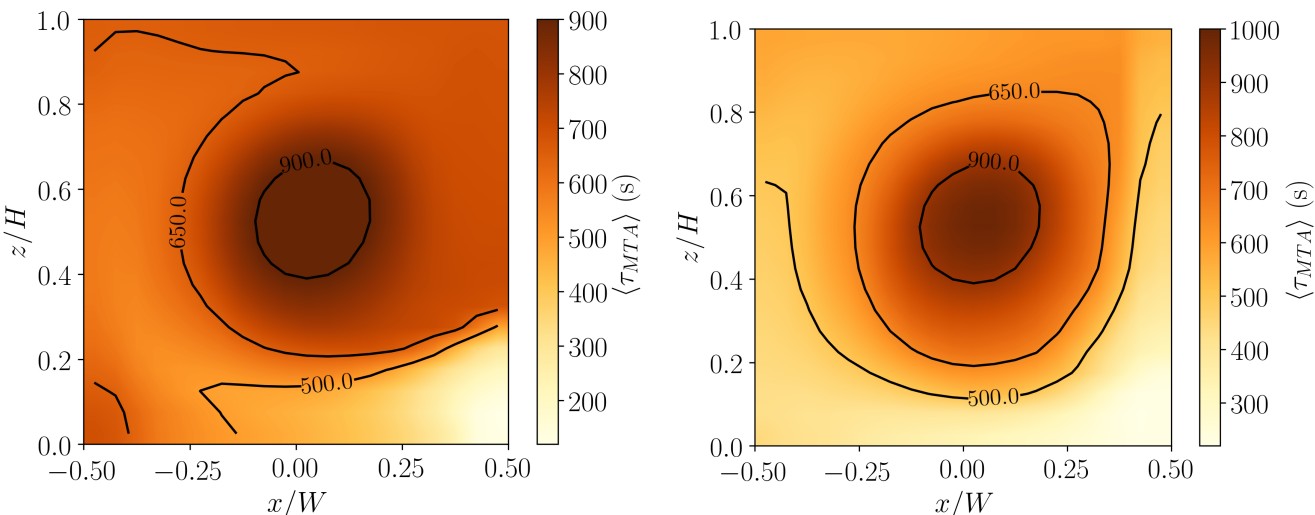

**Figure 14.** Spanwise-averaged MTA for (a) near-ground; (b) column source. The canyon-averaged MTAs are similar (652 s for the NG source and 599 s for the CO source); however, the spatial structures are noticeably different.

To test these predictions, the age may be calculated by tracking fluid parcels. The mean tracer age (MTA; Lo and Ngan, 2015) characterises the average time elapsed between the release of a pollutant at the source and its arrival at the receptor. Following the procedure reviewed in Appendix F, the MTA is calculated for a near-ground source (Fig. 14a). The MTA is lowest near the source at the bottom windward corner and increases towards the centre of the domain, where there are very high values. This pattern is seen most clearly in $RD_{COAG}$ (Fig. 10d) and to a lesser extent, $RD_{ALL}$ (not shown). Linear

correlation coefficients between the MTA and the relative difference for NG-D confirm that coagulation is strongly dependent on the age (Table 4). The correlation is weaker for deposition because the MTA is not maximised around the leeward corner. The magnitude of the correlation with $RD_{DEPO}$ is comparable to that with the total number concentration (not shown), which





tends to decrease away from the source. For a column source (Fig. 14b), the correlation between the MTA and the relative differences is comparable.

| MTA | $RD_{COAG}$ | $RD_{DEPO}$ | $RD_{ALL}$ |
|---|---|---|---|
| NG-D | 0.90 | 0.57 | 0.77 |
| CO-B | 0.86 | -0.15 | 0.16 |

**Table 4.** Linear correlations between the relative differences and the MTAs for NG-D and CO-B.

The preceding results imply that resolving the transient dynamics is potentially important for accurate simulation of aerosol dynamical processes. Since coagulation is nonlinear in the concentration and the concentration evolves between the source and receptor, approximating the coagulation term with a time average will introduce errors. These errors could be significant when coagulation is strong, such as is the case for cooking emissions.

## 5   Sensitivity tests

### 5.1   Background concentrations

The calculations described in Sect. 3 neglect background concentrations, i.e. $N_b = 0$. Although $N_b$ is fixed, the background is still involved in aerosol processes. To assess the effect of the background on the aerosol processes, several cases with $N_b > 0$ are considered.

1. Idealised background spectrum. Here it is assumed that the background spectrum is identical to the emission spectrum.
Using a single emission scenario, NG-B, two cases are considered: (i) light pollution, $N_b = 0.1 N_0$; (ii) heavy pollution, $N_b = 0.4 N_0$, where $N_0$ is the mean canyon-averaged number concentration for $N_b = 0$. These values are arbitrary; however, the increase in $N_b$ is meant to capture the contrast between normal conditions and a severe pollution episode.

2. Realistic urban background spectrum. Here NG-B is applied to the background spectrum from a real urban measurement (Fig. 15d). The measurement was taken in Tsuen Wan, Hong Kong (see Appendix G for details). The measured spectrum
resembles the emission spectrum for traffic (Fig. 3). The measurements suggest $N_b = 0.5 N_0$.

| Baseline | Idealised | | Realistic |
|---|---|---|---|
| $N_b = 0$ | $N_b = 0.1 N_0$ | $N_b = 0.4 N_0$ | $N_b = 0.5 N_0$ |
| 16.96% | 16.03% | 11.41% | 10.69% |

**Table 5.** Canyon-averaged $RD_{ALL}$ for NG-B and different background cases. Note that the baseline concentration ($N_b = 0$) is identical for idealised and realistic cases.





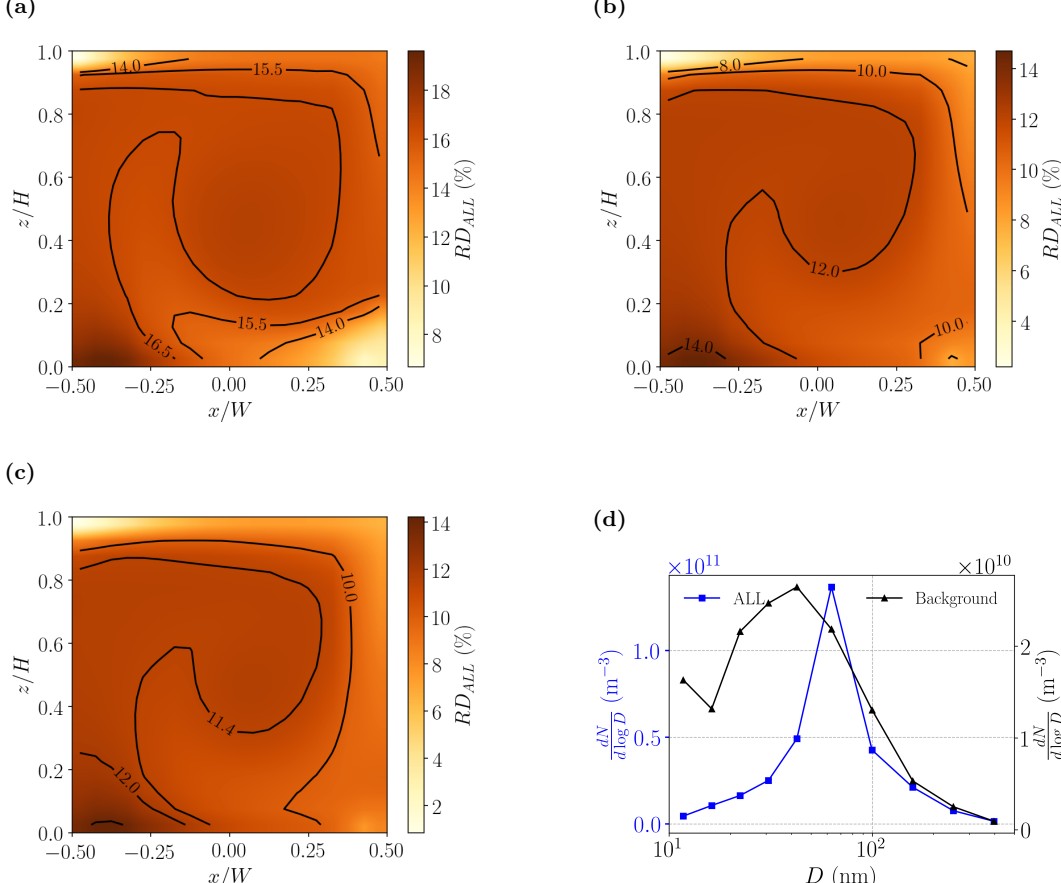

**Figure 15.** Spanwise-averaged relative difference fields of aerosol dynamical processes (ALL). (a) light pollution; (b) heavy pollution; (c) realistic case; (d) aersol number size distributions for background concentration (black line) and total concentration (blue line).

The effect of the background is assessed by calculating $RD_{ALL}$ for different $N_b$. For the idealised cases, $RD_{ALL}$ decreases as the background concentration is increased to $N_b = 0.1N_0$ and $N_b = 0.4N_0$ (Table 5); for the realistic case, $RD_{ALL}$ decreases by up to 37% from its value for $N_b = 0$. Nevertheless, the actual impact is smaller than it may appear on first glance. In the absence of any additional aerosol processes, e.g. if the background were completely inert, $RD_{ALL}$ would also decrease

from the accompanying increase in the total number concentration, $N$: for $N_b = 0.1N_0, 0.4N_0$ and $0.5N_0$, the decreases would be 15.4%, 12.1% and 11.3%, respectively. The values in Table 5 depart from the crude linear scaling, but the deviations are small, i.e. $\sim 5\%$. The qualitative similarity of the $RD_{ALL}$ fields (Figs. 15a-c) suggests that the physical mechanisms described in Sec. 4 are robust. This is true even though the simulated size distribution shifts towards large particles when the realistic background spectrum is used (Fig. 15d).

The effect of the background on $N$ is relatively small because $N_b$ is fixed. Hence aerosol processes involving the background do not change $N$ directly: the background affects $N$ only indirectly through the emissions. For example, coagulation between





the background and emissions will change the size distribution of the nominal emissions (i.e. the "perturbation" or $N - N_b$), which will in turn affect the deposition and coagulation rates. This is clearly a nonlinear process.

## 5.2 Source flux

The dependence on the source flux is usually ignored in studies of pollutant dispersion. For a passive scalar emitted by a uniform source, the concentration scales linearly with the source flux and this dependence can be removed by the nondimensionalisation, eq. (1). For aerosols, however, this is no longer true because coagulation depends nonlinearly on the number concentration. This is potentially important because cooking emissions are not constant with time. Here we assess the sensitivity to the source flux $Q$ for Case CO-B. Letting the original source flux be denoted by $Q_0$, we now consider $Q = \alpha Q_0$ for $\alpha \in [0.1, 10]$.

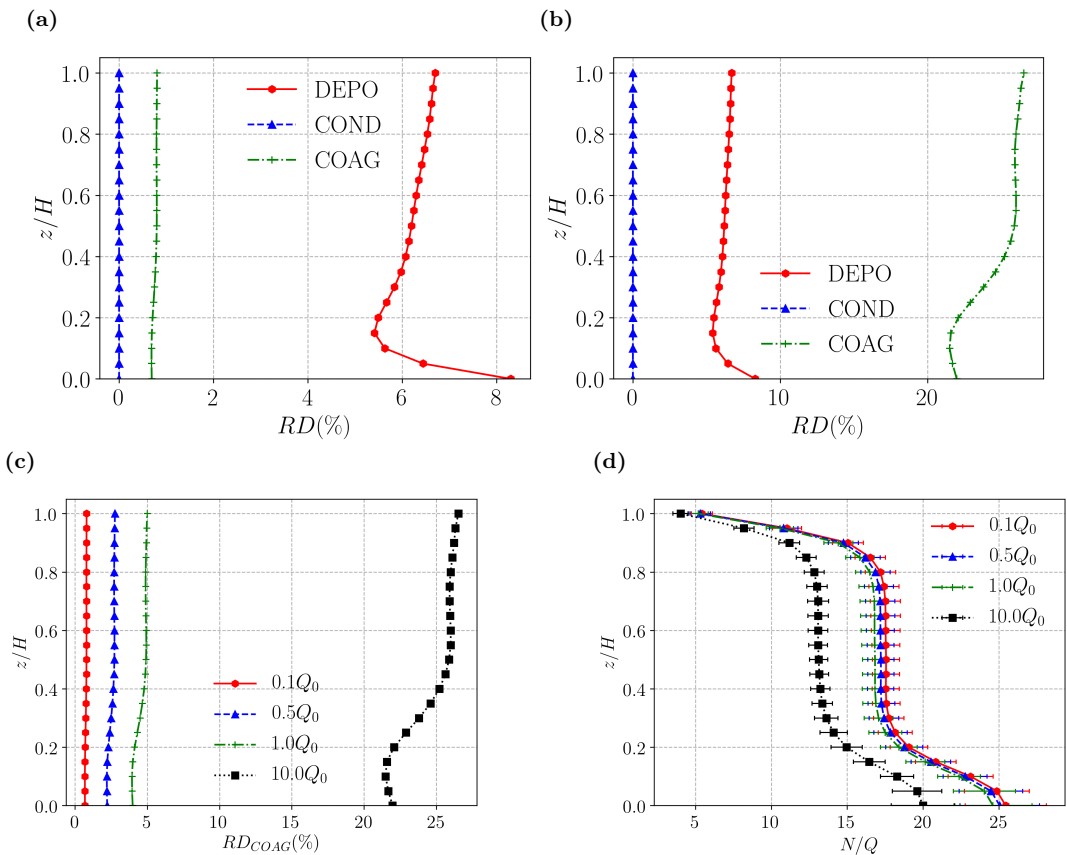

**Figure 16.** Comparison of aerosol processes and source fluxes, $Q$, for Case CO-B, The top panels compare vertical profiles of $RD_i$ at fixed $Q$, the bottom panels compare the effect of different $Q$ on coagulation. (a) $Q = 0.1Q_0$; (b) $Q = 10Q_0$; (c) $RD_{COAG}$; (d) normalised number concentration, $N/Q$.

The effect on the different aerosol processes is illustrated by vertical profiles of $RD_i$. For $Q = 0.1Q_0$ (Fig. 16a) and $Q = 10Q_0$ (Fig. 16b), condensation is negligibly small. Deposition also shows weak sensitivity to the source flux as $RD_{DEPO} \sim$





8% despite the hundredfold increase in $Q$. However, coagulation shows strong sensitivity to $Q_0$: it is small for $Q = 0.1Q_0$ ($RD_{COAG} \sim 0.5\%$) but of major importance for $Q = 10Q_0$ ($RD_{COAG} \sim 25\%$). Examining $RD_{COAG}$ separately (Fig. 16c), the sensitivity is fairly weak for $Q/Q_0 \leq 1$.

The preceding results may be explained as follows. The dry deposition flux for size bin $i$ is linearly proportional to the concentration, i.e. $F_{d,i} = -v_d n_i$, where $v_d$ is the deposition velocity (Seinfeld and Pandis, 2016). The number concentration is directly proportional to $Q$ but this dependence disappears from $RD_{DEPO}$ after the nondimensionalisation. The corresponding coagulation flux (i.e. the contribution of coagulation to the time evolution of $n_i$) is quadratic in $n_i$, implying that coagulation should depend sensitively on the source flux. This is partially confirmed by vertical profiles of the normalised concentration,
$N/Q$ (Fig. 16d), which suggest a nonlinear dependence on $Q$ for $Q/Q_0 > 1$.

## 6   Discussion

This study has focused on an idealised flow and a small number of representative emission scenarios in order to highlight the basic processes governing the dynamics of cooking-generated urban aerosols. We now discuss how the results may extend to more realistic cases.

The emission scenarios defined in Table 1 are arbitrary. Obviously other kitchens dimensions or locations could be chosen, but the results presented here should serve as a starting point for future studies of specific urban environments. The sensitivity to the source flux cannot be strictly neglected, but at least for deviations of $\sim 50\%$ from the baseline value, the effect is modest (Fig. 16). The qualitative response to the source location can be estimated from the behaviour for a passive scalar as the inclusion of aerosol processes has little effect on spatial structure of the number concentration fields (e.g. Fig. 1b). In practice,
the specification of the emission spectrum is a greater concern: mean concentrations for boiling and deep frying differ by $\sim 30\%$ for near-ground emissions and $\sim 15\%$ for column emissions (Table 2). The sensitivity to the emission spectrum would need to be examined on a case-by-case basis.

     A highly simplified representation of indoor aerosol processes was adopted. Since details of the kitchen ventilation systems and exhaust ducts vary, we focused on a configuration in which indoor aerosol processes and the effects of the ventilation system
are ignored: it is assumed that all cooking-generated particles escape without loss, modification or re-entrainment. The results described in this study therefore correspond to an idealised but important limit. Given the fairly weak sensitivity to source flux and source location, one may expect that the results will not be greatly affected by the indoor aerosol processes unless there is a significant change in the emission strength or spectrum. The characteristic timescales for the outdoor aerosol processes (Fig. 13) suggest that the effects of indoor coagulation and deposition could be relatively weak; however, detailed analysis
of ventilation systems would be required to quantify the effect. Calculations including an urban background derived from real urban measurements show that the spatial structure of the aerosol processes is essentially independent of the background spectrum. Moreover, the quantitative effect on the relative differences is small compared to the change in the total particle number.





The restriction to flow over a unit-aspect-ratio street canyon represents a different kind of idealisation. With a building array or realistic urban topography, the spatial inhomogeneity of the flow increases and pollutants may disperse through intersections as well as across the roof level; nevertheless, the qualitative conclusions about the relative importance of deposition and coagulation should generalise for a fairly broad range of conditions. The analysis of Sect. 4 suggests that, for a given source type, the spatial structure of deposition and coagulation are determined primarily by two factors: the streamline geometry or mean circulation and the ratio of the aerosol and dynamical timescales. Deposition is promoted when particles are brought into repeated contact with solid surfaces, such as occurs within corner vortices. Coagulation depends on the ageing of pollutants, which occurs along fluid trajectories in the outdoor environment. With more realistic flows, vortex-like structures are ubiquitous because vortices form in the lee of obstacles. So long as the background winds are unchanged, dynamical timescales and ventilation statistics should not be significantly affected (e.g. Lau et al., 2020). Changes, however, may be expected for unstable stratification, in which recirculations are replaced by convective plumes and ventilation timescales are much shorter (Duan and Ngan, 2020). Preliminary calculations in which the effect of local heating on the emissions is considered indicate that the effect can be quite large (e.g. mean concentrations decrease substantially when the entire surface of a column source is heated; not shown).

The assumptions described above may partially explain why the mass concentrations (Appendix D) are very high. For example, concentrations would decrease if the assumption of perfect ventilation to the exterior were relaxed.

## 7   Conclusions

Cooking-generated aerosols differ significantly from traffic-generated ones. Using standard emission spectra and plausible assumptions about the traffic volume and restaurant dimensions, it was found that the number concentration within a unit-aspect-ratio street canyon is $\sim 50 - 100\%$ higher for boiling and deep frying than for traffic. This reflects differences in the emission factors and the increased importance of deposition and coagulation. The latter is especially important for deep frying, which generates many small particles with diameter $D_p < 50$ nm. The results support the finding that organic aerosols may be determined primarily by cooking emissions in neighbourhoods with many restaurants (Lee et al., 2015; Liu et al., 2018). Even larger differences are seen in the mass concentration, though the effect of aerosol processes on PM2.5 is much smaller.

The sensitivity of the results to the source spectrum and source location can be understood by analysing the deposition and coagulation timescales. For the different cooking-emission scenarios, both timescales are comparable to or longer than the characteristic timescale for large-scale motions within the canyon. The upshot is that deposition is enhanced within corner vortices while coagulation occurs following fluid trajectories. The mean tracer age, which characterises the ageing of particles released at the source, reveals the spatial structure of coagulation.

The present study is restricted to idealised flow and emission scenarios. It would be instructive to perform a similar study for a real street canyon that contains restaurants. This would enable the impact of assumptions about the kitchen emission factors or the neglect of heated plumes to be assessed. However, in situ measurements, preferably of the size spectrum, would be required.





## Appendix A: Aerosol parameterisations

### A.1 Deposition

Dry deposition occurs when particles impact and stick to a solid surface. Many schemes have been developed for calculating
the dry deposition velocity, $v_d$. In SALSA, the scheme of Zhang et al. (2001) is applied:

$$v_d = v_g + \frac{1}{R_a + R_s}, \tag{A1a}$$

$$u_g = \frac{\rho d_p^2 g C}{18\eta}, \tag{A1b}$$

$$R_a = \frac{\ln z_R/z_0 - \psi_H}{\kappa u_*}, \tag{A1c}$$

$$\frac{1}{R_s} = \varepsilon_0 u_* \left( Sc^{-\gamma} + \left( \frac{St}{0.8+St} \right)^2 + \frac{1}{2} \left( \frac{d_p}{A} \right)^2 \right) R_1, \tag{A1d}$$

where $v_g$ is the gravitational settling velocity; $R_a$ and $R_s$ represent the aerodynamic resistance and surface resistance, respectively; $\rho$ is the particle density; $g$ is the acceleration of gravity; $\eta$ is the viscosity; and $C$ is a correction factor. In Eq. A1c, $z_R$ is the height, $z_0$ is the roughness length, $u_*$ is the friction velocity, $\psi_H$ is the stability function, and $\kappa$ is the von Karman constant. For LES, $u_*$ is estimated by $\sqrt{C_D U}$, where $C_D$ is the drag coefficient and $U$ is the local velocity magnitude. $Sc$ is the particle Schmidt number and $St$ is the Stokes number. $\varepsilon_0$, $\gamma$ and $A$ are constants based on the surface type.

### 430 A.2 Coagulation

Coagulation occurs happens when two particles collide and form a larger one. Following Jacobson and Jacobson (2005), the number concentration for size bin $i$ is given by

$$n_{i,t} = \frac{n_{i,t-\Delta t} + \frac{1}{2}\Delta t \sum_{j=1}^{i-1} \beta_{i-j} n_{i-j} n_{j,t-\Delta t}}{1 + \Delta t \sum_{j=1}^{\infty} \beta_{i,j} n_{j,t-\Delta t}} \tag{A2}$$

where $\delta_t$ is the time step and $\beta$ represents the coagulation kernel ($\mathrm{cm^3\,particle^{-1}\,s^{-1}}$) for particles in size bins $i$ and $j$. The
coagulation kernel or rate coefficient is calculated as

$$\beta_{i,j} = \frac{4\pi(r_i+r_j)(D_i+D_j)}{\frac{r_i+r_j}{r_i+r_j+(\delta_i^2+\delta_j^2)^{1/2}} + \frac{4(D_i+D_j)}{(\overline{\nu_{pi}}^2+\overline{\nu_{pj}}^2)^{1/2}(r_i+r_j)}}. \tag{A3}$$

Here, $D$ is the particle diffusion coefficient; $\delta$ denotes the mean distance from the centre of a sphere defined by the particle mean free path, $\lambda_{pi}$; $\overline{\nu_p}$ is the thermal velocity.





## A.3 Condensation

Condensation increases the particle volume through mass transfer. Following Jacobson and Jacobson (2005), the vapour mole concentration of a condensing gas $g$ is calculated as

$$C_{g,t} = \frac{C_{g,t-\Delta t} + \Delta t \sum_{i=1}^{N}(k_{g,i,t-\Delta t}S'_{g,i,t-\Delta t}C_{g,s,i,t-\Delta t})}{1 + \Delta t \sum_{i=1}^{N} k_{g,i,t-\Delta t}}, \tag{A4}$$

whence the particle mole concentration may be updated,

$$c_{g,i,t} = c_{g,i,t-\Delta t} + \Delta t k_{g,i,t-\Delta t}(C_{g,t} - S'_{g,i,t-\Delta t}C_{g,s,i,t-\Delta t}). \tag{A5}$$

Here, $k_{g,i,t-\Delta t}$ is the mass-transfer coefficient $(\mathrm{s}^{-1})$; $S'_{g,i,t-\Delta t}$ is the equilibrium saturation ratio; and $C_{g,s,i,t-\Delta t}$ is the uncorrected saturation vapour mole concentration $(\mathrm{mol\,m}^{-3})$.

## Appendix B: Chemical compositions and emission factors for gaseous compounds

**Table B-1.** Chemical compositions for cooking (See and Balasubramanian, 2008) and traffic (Yubero et al., 2015) emissions. In the former case, a gas stove is assumed.

|  | Composition(%) | Mass ($\mu g\,m^{-3}$) | OC | BC | Cl– | $NO_3^-$ | $SO_4^-$ | $NH_4^+$ |
|---|---|---|---|---|---|---|---|---|
| Cooking | Deep-frying | $82.3 \pm 40.8$ | 60.8 | 7.3 | 0.21 | 3.5 | 0.5 | 0.3 |
|  | Boiling | $40.9 \pm 11.8$ | 43.0 | 8.5 | 2.9 | 7.3 | 1.8 | 0.3 |
|  | Traffic |  | 43.0 | 17.5 | 0.0 | 5.0 | 24.8 | 9.7 |

**Table B-2.** Gaseous emission factors for cooking (Shen et al., 2018) and traffic (Kumar et al., 2008) emissions. In the latter case, the emission factors are derived by assuming that that there is one stove for a kitchen of volume 16 m³.

|  | $H_2SO_4$ | $HNO_3$ | $NH_3$ | NVOC | SVOC |
|---|---|---|---|---|---|
|  |  |  | $\mathrm{g\,km^{-1}\,veh^{-1}}$ |  |  |
| Traffic | $2.5 \times 10^{-4}$ | 0.0 | $4.2 \times 10^{-2}$ | 0.0 | $2.5 \times 10^{-3}$ |
|  |  |  | $\mathrm{g\,min^{-1}}$ |  |  |
| Cooking | 0.0 | $3.8 \times 10^{-3}$ | 0.0 | 0.0 | $5.5 \times 10^{-4}$ |





**Appendix C: Validation of scalar statistics**

Time-averaged concentration statistics are compared with the wind-tunnel data of Pavageau and Schatzmann (1999) by intro-
ducing a ground-level line source along the central axis of the unit-aspect-ratio street canyon. The numerical configuration is
otherwise unchanged from Sect. 2.3. Fig. C-1 shows normalised concentration profiles at different streamwise positions. Con-
centrations are consistently overpredicted at the leeward wall, centre and windward wall; however, the agreement is at least as
good as the previous LES validation of Michioka et al. (2011). In order to quantify the agreement, standard air quality metrics
(Chang and Hanna, 2004) are calculated:

$$\mathrm{FB} = \frac{\overline{C_o} - \overline{C_p}}{0.5(\overline{C_o} + \overline{C_p})}, \tag{C1}$$

$$\mathrm{NMSE} = \frac{\overline{(C_o - C_p)^2}}{\overline{C_o}\,\overline{C_p}}, \tag{C2}$$

where $C_p$ and $C_o$ denote the model predictions and observations, respectively. A perfect model would have $\mathrm{FB} = \mathrm{NMSE} = 0$.
For the validation, $NSME = 0.07$ and $FB = 0.2$, indicating relatively good agreement. We conclude that the model is capable
of predicting mean concentrations for a passive scalar within a street canyon.





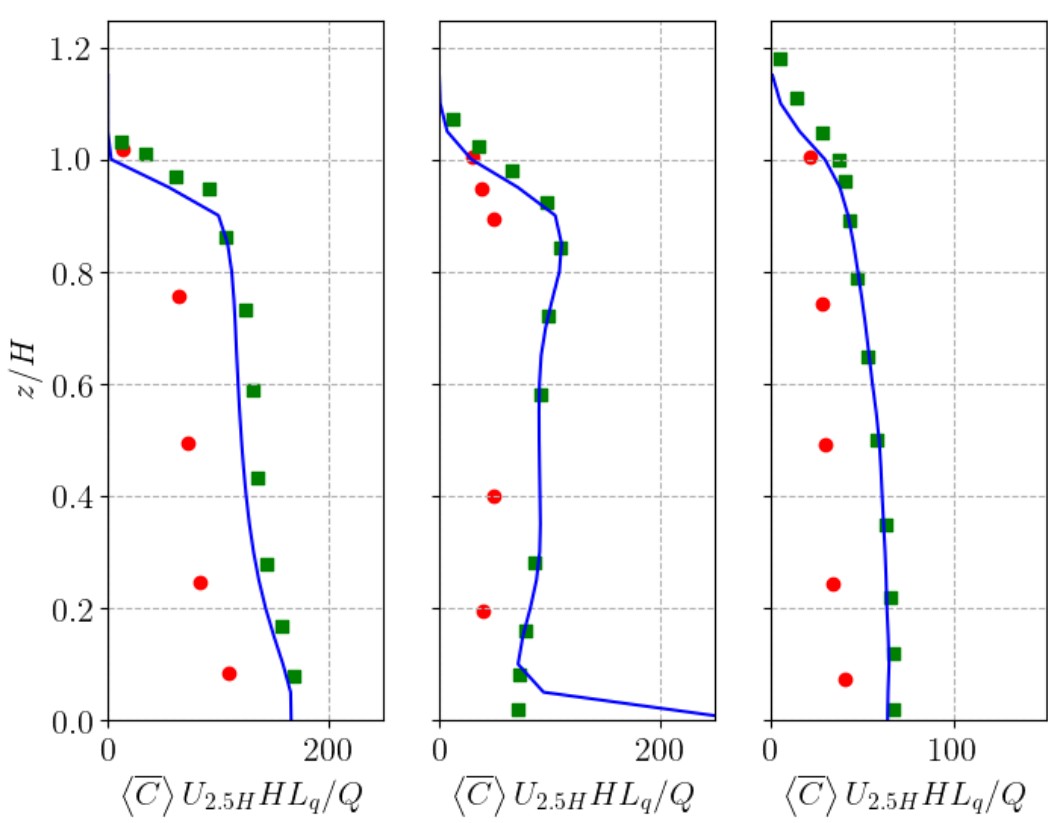

**Figure C-1.** Vertical profiles of the normalised mean concentration plotted at $x/W$ = -0.5, 0 and 0.5. The results from present LES simulations are plotted in blue lines and are compared with wind-tunnel data from Pavageau and Schatzmann (1999) (red circles) and LES data from Michioka et al. (2011) (green squares). $U_{2.5H}$ denotes the the temporal and spatial average of the streamwise velocity at $z/H = 2.5$ and $Q$ is the source flux.





**Appendix D: Mass concentrations**

Mean mass concentration fields for traffic and near-ground emissions are plotted in Fig. D-1.

**Figure D-1.** As in Fig. 6, but for PM2.5 concentrations. The mass concentrations are largely insensitive to aerosol dynamic processes.





## Appendix E: Comparison of aerosol dynamical processes for other emission scenarios

Results corresponding to Figs. 10 and 11, but for Cases NG-B and CO-B are now shown.

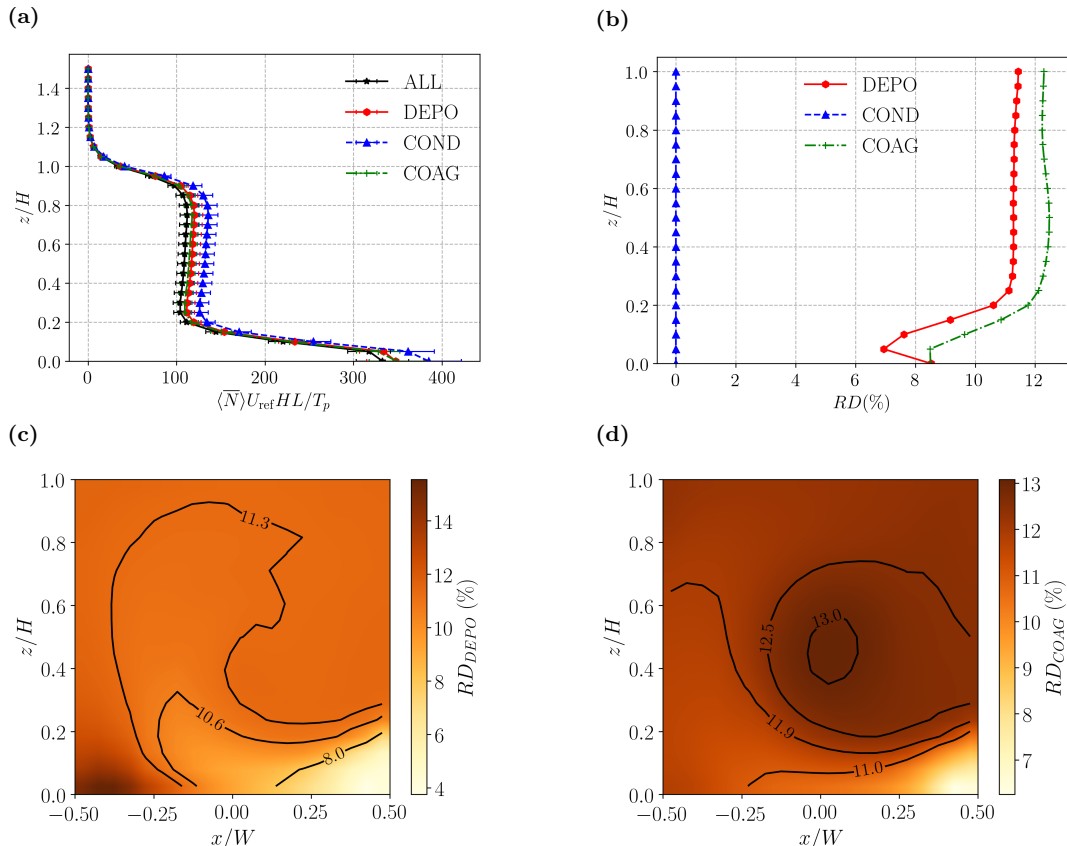

**Figure E-1.** As in Fig. 9, but for Case NG-B.

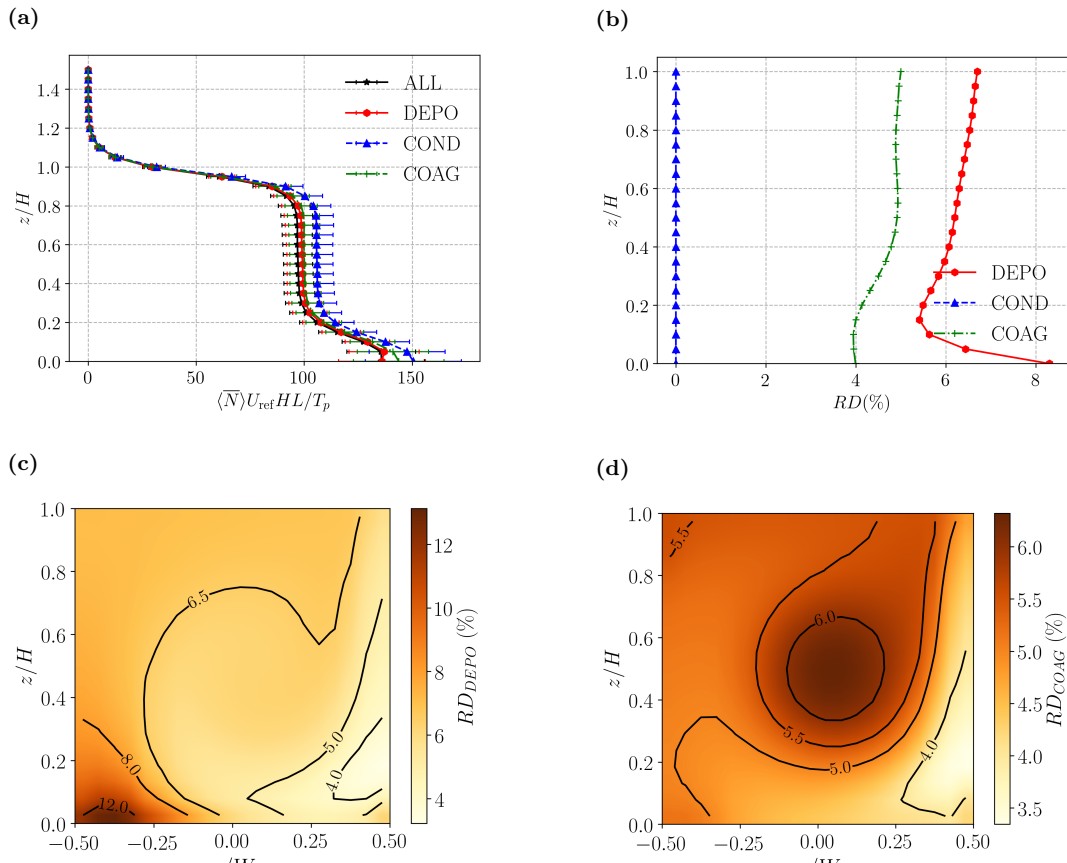

**Figure E-2.** As in Fig. 9, but for case CO-B.





**Appendix F: Calculation of the mean tracer age**

The mean tracer age measures the time elapsed from the release of a passive scalar at a source location to its arrival at the receptor. The theory is described in Holzer and Hall (2000) and Lo and Ngan (2015). Briefly, a Green's function, $G(\mathbf{x}|\mathbf{x}_0)$, which maps the scalar concentration from the source $\mathbf{x}_0$ to the receptor $\mathbf{x}$, is obtained from the solution of the advection-diffusion equation for an impulse source (i.e. delta function in time). The age spectrum or probability distribution, $Z$, of transit times, $\xi$, is given by

$$Z(\mathbf{x},\xi) = \frac{\int_D G(\mathbf{x},\xi|\mathbf{x_0})S(\mathbf{x_0})d\mathbf{x_0}}{c(\mathbf{x},t)}, \tag{F1}$$

where $S$ refers to the source and $c$ is the concentration. The mean tracer age is the first moment of the age spectrum, i.e.

$$\tau_{MTA} = \int_0^\infty \xi Z d\xi. \tag{F2}$$





## Appendix G: Measurements of the background size spectrum

The measurements were conducted from 1 to 8 March 13:00-18:00 local time on the roof of Hoi Pa Street Government Primary
School, Tsuen Wan, Hong Kong (height: 31 m, coordinates: 22.372° N, 114.115°E, Fig. G-1). Using a Kanomax PAMS 3300 spectrometer and 14 fixed channels, the number distribution was measured from 14.51 nm to 862.32 nm. The number spectra (75 in total) were averaged to yield the spectrum of Fig. 15d. The site is located in a suburban neighbourhood. During the measurement period, the average traffic volume along the main road (Tai Ho Road) was 2050 veh/h. The mean number concentration, $\bar{N} = 2.3 \times 10^{10} \, \mathrm{m^{-3}}$, implying $N_b = 0.5N_0$, where $N_0$ is the mean canyon-averaged number concentration for
NG-B.

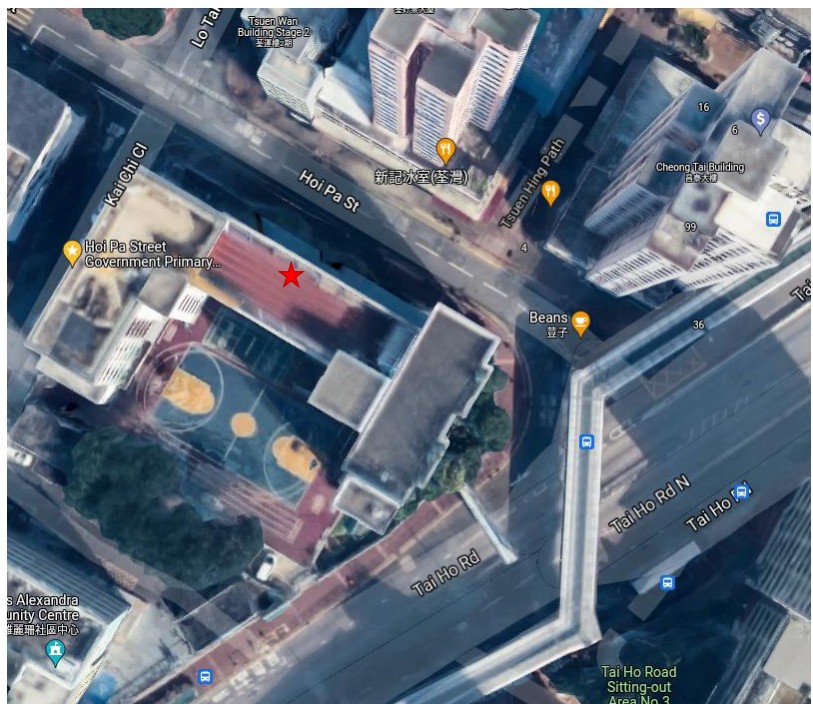

**Figure G-1.** Measurement site in Tseun Wan, Hong Kong (image taken from © Google Maps). The measurement location is indicated by the red star.

*Code availability.* The codes used in this publication are available to the community, and they can be accessed by request to the corresponding author.



*Author contributions.* SG conducted the simulations with MK developing the model code. SG analysed the data. SG, KN and CKC wrote the paper. All co-authors contributed to the discussion of the paper.

*Competing interests.* The authors declare that they have no conflict of interest.

*Acknowledgements.* Financial support was provided by the Environmental and Conservation Fund (Project 7/2020), Guangzhou Development District International Science and Technology Cooperation Project (No. 2018GH08), and City University of Hong Kong (Project 7005283). The authors thank Alvin CK Lai for lending the portable spectrometer.



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
