# Peer review of "Technical note: Dispersion of cooking-generated aerosols from an urban street canyon"

_Atmospheric Chemistry and Physics, 2021_

## Author Comment (AC1)

**Response to Referee 1**

We thank the referee for the helpful comments. Point-by-point responses are included below. Briefly we clarify the main findings of the ms and include some additional sensitivity calculations.

**General comments**

**Comment**

The manuscript deals with the interesting and atmospheric relevant topic of the dispersion and processing of cooking and road traffic generated aerosol particles in urban street canyons. The authors address this topic using the building-resolving computational fluid dynamics model PALM, which also include the sectional aerosol dynamics module SALSA. The model was setup for an hypothetical and simplified street canyon with road traffic emissions or cooking generated aerosol emissions from the surrounding buildings. The authors address how the type, location and aerosol dynamics of the emissions influence the concentration in the street canyon.

Apart from a few typos and minor grammatical errors I think the manuscript is generally well written. I agree with reviewer 2 that it could benefit from some restructuring. Also consider if you need all 16 figures. After careful revision I think the manuscript has the potential to be publishing as an atmospheric relevant "technical note" in atmospheric chemistry and physics.

**Response:** Yes, we agree that there were too many figures. We have moved two figures from Appendix E to Supplementary Material. New figures and tables have also been moved to Supplementary Material.

**Comment**

What I mainly miss with the manuscript is a more careful motivation to the choice of the simplified (idealized) street canyon, the primary particle emission size distributions from the different emission sources, the meteorological conditions and the location of the cooking emissions, especially since the model is compared against real observations of wind velocity profiles from a wind-tunnel study (Figure 4), and observations from a specific street canyon in Cambridge (Figure 5).

**Response:** Yes, this is a good point. In any numerical study of urban pollutant dispersion, the representativeness of the geometry, meteorological conditions and source specification is always an issue. The choices made in this manuscript are generic ones of wide applicability:

The street canyon is recognised as the basic geometric unit of the built environment (Oke 1988). A unit aspect ratio is a canonical choice because it mimics the effect of deep urban canopies, where relatively poor ventilation and strong pollutant trapping occur, though of course the precise flow details differ. The simplicity of the street canyon geometry makes it especially suitable for investigating the effects of physical processes such as chemistry (e.g. Zhong et al. 2015).

The assumption of constant density and a wind perpendicular to the canyon axis is another standard choice that is of great relevance to urban air quality. Ventilation improves for unstable conditions and is largely unchanged for stable stratification (Duan & Ngan 2019; see also our reply to Referee 2). With respect to the wind direction, a perpendicular external wind leads to the occurrence of a canyon vortex and strong pollutant trapping.

The source locations and emission spectra are also intended to be generic choices. Just as with vehicular emissions, in which moving discrete sources are represented by a line or area source, the near-ground, isolated and column cooking sources are meant to be plausible idealisations. A systematic derivation lies far beyond the scope of this study: indeed, we are unaware of a comparable analysis for vehicular emissions. The emission spectra are taken from well-known studies.

The generic nature of these choices is now mentioned in the Introduction (l. 45):

**After reviewing the methodology (Sect. 2), results are presented for several idealised but generic emission scenarios, e.g. traffic, deep frying and cooking emissions (Sect. 3).**

We also argue in the Discussion that these choices do not affect one of the key findings of this study, which is that the nature of the aerosol dynamics is largely determined by the ratio of the coagulation and deposition timescales to the dynamical timescale. Of course, fine details of the aerosol dynamics necessarily depend on the choices described above; however, the relative importance of coagulation and deposition is largely insensitive to most of these choices. In the case of cooking emissions, similar behaviour is obtained so long as the coagulation timescale ($\tau_{coag}$) remains long compared to the deposition ($\tau_{depo}$) and dynamical timescales. The dynamical timescale, which may be taken to be the mean canyon circulation timescale ($T_c$) or the mean tracer age (MTA), characterises the time required for a pollutant to escape from the canyon or the amount of time elapsed since the pollutant was released. This argument is closely related to that of Harrison 2018, who showed that the dynamics of gas phase pollutants depends on the ratio of the chemical timescale to the dynamical timescale (residence time). Roughly speaking, the extent of chemical transformation within the canopy depends on the timescale over which pollutants are allowed to react. In the present case, the relevant dynamical timescale for coagulation is the total time elapsed since a particle was emitted. Small changes in $\tau_{depo}$ or $\tau_{coag}$, which inevitably accompany modifications to the configuration, should not have a significant effect on the aerosol dynamics.

We have added a new table (Table 5) that summarises these timescales for the different emission scenarios.

**Table 5.** Dynamical and aerosol timescales for different emission scenarios.

| Source location | $T_c$ (s) | MTA (s) | Emission scenario | $\tau_{depo}$ (s) | $\tau_{coag}$ (s) |
|---|---|---|---|---|---|
| TR | 382 | 584 | TR | 150 | $1.1 \times 10^7$ |
| NG | 382 | 652 | NG-D | 150 | $1.8 \times 10^5$ |
|  |  |  | NG-B | 150 | $1.2 \times 10^6$ |
| CO | 382 | 599 | CO-D | 150 | $2.2 \times 10^5$ |
|  |  |  | CO-B | 150 | $1.8 \times 10^6$ |

In all cases, $\tau_{depo}$/MTA <1 and $\tau_{coag}$/MTA >>1. We also discuss how the ratio of these timescales may change for different configurations (Sec. 6, l.372):

> **With other emission scenarios or flows, quantitative differences are unavoidable, but qualitative differences in the aerosol dynamics are not expected in most cases. For cooking emissions, the coagulation timescale is much longer than the relevant dynamical timescale (Table 5), which implies that coagulation will continue to be controlled by the ageing of fluid parcels or the mean tracer age. The dynamical timescales change with the wind direction (Supplementary Material, Table S-1), but the coagulation timescale, $\tau_{coag}$, remains much longer. For stratified flow, the MTA will decrease for unstable stratification and increase for stable stratification but the effect should be relatively small (see Duan and Ngan (2019) for building array results). The situation is more complicated for deposition insofar as $\tau_{depo}$ is not much less than the relevant dynamical timescale, i.e., the canyon circulation timescale $T_c$. Qualitatively different behaviour is expected only for a much smaller $T_c$, such as may occur for unstable flow or a street canyon with lateral openings. In this case, deposition will be less spatially localised and will no longer proceed to completion. For cooking emissions, the relative contribution of deposition would therefore decrease compared to the cases examined in this paper, for which $\tau_{depo}$/$T_c$ < 1.**

It is likely that the coagulation timescale will remain long in most cases. The deposition timescale may not be short relative to the dynamical timescale for certain cases, e.g. convectively unstable flows or a finite street canyon, but this should serve to reinforce the importance of coagulation for cooking emissions.

**Comment**

No very much specific information is given about how the aerosol dynamics is represented in the model in the current study. Only the primary particle emissions are described, with some details. Especially I miss information about how the condensation of different vapors were treated in the model. E.g. what properties were assigned to the semi-volatile condensable vapors HNO3, NH3 and SVOCs.and how do you calculate their volatility with respect to the aerosol particle phase? For HNO3, NH3 is should depend on the aerosol liquid water content and acidity

**Response:** The treatment of condensation is described in Appendix A. Since chemical transformations are not considered in this study, we focus on the equilibrium saturation ratio and saturation vapour mole concentration. Other properties, such as the volatility with respect to the aerosol particle phase, are therefore excluded.

**Specific comments**

Abstract, I miss one sentence which motivate why this study is important from an atmospheric chemistry and physics perspective.

**Response:** A new sentence summarising the points made in response to the general comments has been added to end of the Abstract:

**It is argued that the qualitative nature of the aerosol dynamics within urban canopies is determined by the ratio of the aerosol timescales to the relevant dynamical timescale (e.g. the mean age of air).**

L24-26, "Deposition is usually the only aerosol process included in urban CFD models as it is the most important for traffic emissions within street canyons (Kumar et al., 2011)." The reference to this statement is a bit old, is this statement still true? Also consider to replace "as it is the most important" with as it is often assumed to be the most important loss process of ultrafine particles. If you do not consider other process you cannot judge their importance. Hear you also only refer to loss processes and not formation processes such as atmospheric new particle formation which can be a major source of ultrafine particles also in urban environments.

**Response:** The reference is quite old, but the statement is still true. Karl et al 2016 also find that the dry deposition is the most important aerosol process in urban environments (with a relative difference in particle number of ~15%). Furthermore, more recent numerical studies, e.g. Kim et al 2019, only include deposition. We have modified the text and added a reference (l. 25).

**[...] as it is often taken to be the most important loss process of ultrafine particles emitted by [...]**

Line 35-36, "There are strong reasons for expecting the dispersion of traffic-generated and cooking-generated aerosols to differ qualitatively." Consider to reformulate this sentence.

**Response:** The text has been rephrased as follows (l. 35): "The dispersion of cooking- and traffic-generated aerosols differ in two key respects."

L37, "diameter of O(10 nm)" What do you mean with O?

**Response:** O stands for "order of". This is the so-called Big O notation which is commonly used in numerical modelling. The sentence in question notes that cooking emissions contain a higher proportion of particles with a diameter of around 10 nm.

> L44 "The effects on the aerosol dynamic processes are highlighted" Do you mean the effects of the aerosol dynamic processes are highlighted ?

**Response:** No, we did indeed mean to refer to the effects of the emission scenarios *on* the aerosol dynamic processes. The analysis in Sec. 3.3, for example, focuses on how the effects of coagulation and deposition depend on the emission scenario. While the effects *of* the aerosol dynamic processes are obviously related, the emphasis is somewhat different. Nevertheless, the sentence has been deleted for brevity.

> L56-57, "the inclusion of transient dynamics allows for nonlinear aerosol processes to be represented more accurately (see Sec. 5.2)." What do you mean with this statement?

**Response:** Our intention here was to draw a distinction between steady (RANS) and unsteady (LES) calculations. Even if one is interested only in the time average, the neglect of temporal fluctuations is problematical when there are nonlinear terms because the time average of products of the fluctuations does not vanish. The same argument lies behind all turbulence models. In the present case, the modelling of nonlinear aerosol processes such as coagulation will be less accurate with steady RANS. This is now explained on l.56:

> **With a steady model, temporal fluctuations are neglected, thereby necessitating a turbulence parameterisation for the aerosol dynamics.**

> L60-61, "Nucleation, which is computationally expensive to simulate, is not considered in this work." In which way do you mean that nucleation is computationally expensive to simulate? Usually nucleation is parameterized as a rate only depending on e.g. the H2SO4 concentration, or H2SO4 and NH3. The concentrations of these vapors you anyway have to calculate in the model for the condensation growth.

**Response:** Nucleation is computationally expensive because it occurs on a short timescale. According to Rönkkö et al., the nucleation timescale in the exhaust of a vehicle is $\tau_{\text{nucl}} \sim 0.7$s. To resolve the process accurately, a smaller timestep (and a finer grid) is required; Ketzel and Berkowicz noted that it's simpler to represent nucleation by modifying the representation of the source. Nonetheless, the computational cost is not necessarily prohibitive: Kurppa et al. (2019) noted that the computational cost of nucleation is comparable to that of deposition. We have therefore modified the wording (l.60):

> **Nucleation, which is most relevant in the immediate vicinity of the source (Rönkkö et al., 2007) and can be treated by modifying the emissions (Ketzel and Berkowicz, 2004), is not considered in this work.**

Thank you for bringing this point to our attention.

> L71-72, "semi-volatile (NVOCs) and non-volatile organics (SVOCs)". It should be semivolatile (SVOCs) and non-volatile organics (NVOCs)

**Response:** Fixed.

> L73, "however, chemical transformations are excluded." What exactly do you mean with this statement? Did you not consider any gas-phase chemistry at all? If this is the case, please state this clearly.

**Response:** Yes, we simply meant to say that all gas phase chemical reactions are excluded. This is now stated unambiguously in the revised text (l. 72): "[...] gas phase chemical reactions are excluded."

> L83, "The flow is driven by an external pressure gradient, dp/dx = -0.0006 Pa m−1." I cannot judge if this is a reasonable value. Can you add some information about typical values and a reference?

**Response:** This value has been used in many previous studies (e.g. Duan et al 2019). Using this value, the streamwise velocity at z/H = 2.5 is U~3 m/s. This is now explained on l.83:

> **[...] This value has been widely used in previous CFD studies (e.g. Duan et al., 2019); it yields a streamwise velocity U $\sim$ 3ms$^{-1}$ 90 at z/H = 2.5. [...]**

> L109-110, "The emission factor for the number of particles emitted by a vehicle per unit distance travelled is $3.0 \times 10^{14}$ km−1 veh−1 (Fujitani et al., 2020)" This, cannot always be a fixed value. At least replace "is" with e.g. "was estimated to be".

**Response:** Yes, the recommended text is now used.

> L115-116, "The emission factors for the number of particles emitted per unit time by a kitchen of unit volume are $3.75 \times 10^{10}$ m−3 s−1 and $4.31 \times 10^{9}$ m−3 s−1, for deep frying and boiling, respectively." Replace "are" with e.g. "were estimated to be".

**Response:** Yes, "were estimated to be" is now used.

> Figure 3. The selected traffic emission spectrum from Janhäll et al., 2004 is relatively old. Is this still representative for the more modern car fleet today? I imagine that the fraction of nucleation mode particles may have gone up while the soot mode may have decreased with more modern cars? But, I may be wrong. Can you find any more recent references to at least compare

with? Quite a lot of your conclusions are based on the selected size distributions of traffic, deep-frying and boiling emission size distributions.

**Response:** Yes, this is a good point. We have checked some more recent references. The number distribution measured by Schneider et al. 2015 is broadly consistent with the spectrum of Janhäll et al., 2004, as may be seen in Figure 1.

[Figure]

**Figure 1**:  Comparison of size spectra measured within street canyons. Data are taken from Schneider et al. 2015 (red) and Janhäll et al., 2004 (blue).

Indeed, the mean particle diameters (i.e. 47.5 nm and 47.9 nm) agree well.

Line 148-149, "Following K19, the coupled PALM-SALSA model is validated against evening measurements of the aerosol number concentration within a real street canyon in Cambridge, UK (Kumar et al., 2008). Can you really evaluate your model results against these observations? How similar are the Cambridge street canyon compared to your idealized street canyon. How does the meteorological conditions during the measurements agree with the neutral conditions with the temperature fixed at 300 K?

**Response:** Yes, we believe that it is fair to compare our model results to the observations. The street canyon geometry in our model (W=H=12 m, L = 167 m) is essentially identical to that of the real one (W=11.75m, H=11.6 m, L = 167 m). The emissions along the street canyon and background concentrations are identical to those in Kurppa et al. 2019. The main difference is that our domain is much smaller as we exclude the buildings surrounding the street canyon where the measurements were taken.

For consistency with the evening measurements, the temperature was fixed at 274 K for the validation only; the value of 300 K was used for the results proper. This is now explained on l. 149:

**For consistency with the evening measurements, the temperature is fixed at 274 K.**

L151, "only are considered." Change to are only considered.

**Response:** We believe that the original wording, "Using the traffic data in K19, emissions from the street canyon only are considered", more accurately conveys the intended meaning, which is that emissions from neighbouring streets are excluded.

Figure 5, I miss a describing text and reference to Fig. 5 in the manuscript

**Response:** Yes, Fig. 5 is now referred to on l. 151:

**[...] Vertical profiles of the aerosol number concentration are compared in Fig. 5. [...]**

L169-170 "Deep frying (NG-D) and boiling (NG-B) yield identical concentrations in the absence of aerosol dynamic processes. Please explain why this is the case. E.g. Deep frying (NG-D) and boiling (NG-B) yield identical concentrations in the absence of aerosol dynamic processes because the location of the emission sources are identical.

**Response:** When aerosol dynamical processes are excluded, the aerosol evolution for NG-D and NG-B is completely passive. The normalised concentrations (eq. 1) are identical because differences arising from the emission strengths are eliminated. This is now explained on l. 169:

**Deep frying (NG-D) and boiling (NG-B) yield identical normalised concentrations in the absence of aerosol dynamic processes.**

L175-176 "One possible explanation for this discrepancy is that the emission spectra differ: the mean particle size is larger in the current study, i.e. 47.9 nm rather than 32.7 nm." This again makes me wonder about how representative the selected traffic emission spectrum is.

**Response:** As discussed above (see Figure 1), the traffic emission spectrum of Janhäll et al., 2004 does not appear to be inconsistent with more recent measurements. Nevertheless, we agree that representativeness of the emission spectrum is an important issue for this (or any other study of aerosols in the outdoor environment).

To investigate this issue, we have carried out sensitivity calculations for emission scenario NG-B:
- o Displacement to large scales (LD): the particle diameter of each size bin is doubled
- o Displacement to small scales (SD): the particle diameter of each size bin is halved

Vertical profiles of the mean number concentration are similar for LD and SD (Figure 2). Moreover, the relative difference fields show a similar spatial structure (Figure 3). We therefore conclude that the representativeness of the emission spectrum for boiling emissions should not be a serious issue. Similar behaviour may be expected for deep frying.

These results are now referred to in the Discussion:

**Although the results inevitably depend on the emission spectrum — mean concentrations for boiling and deep frying differ by ∼ 30% for near-ground emissions and ∼ 15% for column emissions (Table 2) — there is no evidence for strong sensitivity. Test calculations in which the emission spectrum for NG-B is scaled by a factor of 2 or 0.5 show limited sensitivity. For example, the vertical profiles show a nearly identical shape with mean concentrations differing by less than 5% with respect to the default emission spectrum (Supplementary Material, Fig. S-4) [here Fig. 2]. Furthermore, the spatial structure of the relative difference fields is almost identical (Supplementary Material, Fig. S-5) [here Fig. 3].**

[Figure]

**Figure 2:** Vertical profiles of the mean number concentration for emission scenario NG-B and all aerosol processes. The vertical profiles correspond to the default emission spectrum (ALL), displacement to large scales (ALL-LD) and displacement to small scales (ALL-SD).

[Figure]

**Figure 3:** Relative difference fields for NG-B: (left) default emission spectrum; (middle) displacement to large scales, LD; (right) displacement to small scales, SD.

L176-177 "This is significant because smaller particles may have a larger deposition velocity" When you refer to small particles I think you mean submircon particles < 1000 nm in diameter. In this, case are not small particles (e.g. ultrafine particles) always having greater deposition velocities than larger >100 nm diameter particles?

**Response:**   Actually, by smaller particles we mean particles whose mean diameter is less than 100 nm. For the deposition parameterisation of SALSA, the deposition velocity increases monotonically for D < 100 nm (Kurppa et al. 2019, Figure 1). Clarification has been added to l.176: "[...] because smaller particles (with a diameter less than 100 nm; K19) may have [...]". For urban surfaces, the deposition velocity is usually (but not necessarily) smaller.

L216, "Condensation has a negligible effect ..." Does this not also depend on the model assumptions/limitations? Also evaporation of semi-volatile species from the fresh exhaust particles could potentially have large influence on the particle number size distribution, especially at the selected high temperature of 300 K. Some recent studies claim that particles can grow very rapidly by nitric acid an ammonia condensation, see e.g:

Wang, M., Kong, W., Marten, R. et al. Rapid growth of new atmospheric particles by nitric acid and ammonia condensation. Nature 581, 184–189 (2020). https://doi.org/10.1038/s41586-020-2270-4.
Could the importance of such claimed rapid growth phenomenon be studied and verified or dismissed using PALM-SALSA?

**Response:**   Yes, it's conceivable that condensation could have a greater effect under certain conditions, but given that its primary effect is to increase particle volume, one may expect the effect on the number concentration to be weak in most cases. Nevertheless, the wording has been qualified (l. 216):

> **Condensation has a negligible effect on the aerosol number concentration, which is consistent with the notion that it primarily serves to increase the volume of particles**

We agree that, under the right conditions, evaporation could have a noticeable effect on the particle number size distribution. But given that numerical studies of aerosols

in the urban environment usually include deposition only, we follow Kurppa et al. (2019) and consider deposition, condensation and nucleation only. Inclusion of evaporation would require a very small timestep.

In theory, PALM-SALSA could be used to investigate the occurrence of the rapid growth phenomenon described in the Nature study; however, the computational cost could be quite high as PALM-SALSA is designed for large scales (e.g. neighbourhoods rather than reaction chambers) and longer timescales. Furthermore, a large number of size bins may be needed to resolve the increase in the particle size.

L275 "O" What do you mean?

**Response:** As explained in the comment to l. 37, O refers to the Big O notation.

**References:**
Duan, G., Jackson, J. G., & Ngan, K. (2019). Scalar mixing in an urban canyon. *Environ. Fluid Mech.*, 19(4), 911-939.

Harrison, R.M., (2018). Urban atmospheric chemistry: A very special case for study. *npj Clim. Atmos. Sci.*, 1(1), .1-5.

Karl, M., Kukkonen, J., Keuken, M. P., Lützenkirchen, S., Pirjola, L., & Hussein, T. (2016). Modeling and measurements of urban aerosol processes on the neighborhood scale in Rotterdam, Oslo and Helsinki. *Atmos. Chem. Phys.*, 16(8), 4817-4835.

Ketzel, M., and R. Berkowicz, (2004): Modelling the fate of ultrafine particles from exhaust pipe to rural background: an analysis of time scales for dilution, coagulation and deposition. *Atmos. Environ.*, **38**, 2639–2652.

Kim, M.J., Park, R.J., Kim, J.J., Park, S.H., Chang, L.S., Lee, D.G. and Choi, J.Y., (2019). Computational fluid dynamics simulation of reactive fine particulate matter in a street canyon. *Atmos. Environ.*, **209**, 54-66.

Kurppa, M., Hellsten, A., Roldin, P., Kokkola, H., Tonttila, J., Auvinen, M., ... & Järvi, L. (2019). Implementation of the sectional aerosol module SALSA2. 0 into the PALM model system 6.0: model development and first evaluation. *Geosci. Model Dev.*, 12(4), 1403-1422.

Lo, K. W., & Ngan, K. (2017). Characterizing ventilation and exposure in street canyons using Lagrangian particles. *J. Appl. Meteorol. Climatol.*, 56(5), 1177-1194.

Oke, T.R., (1988). Street design and urban canopy layer climate. *Energy Build.* 11, 103–113.

Rönkkö, T., A. Virtanen, J. Kannosto, J. Keskinen, M. Lappi, and L. Pirjola, (2007): Nucleation Mode Particles with a Nonvolatile Core in the Exhaust of a Heavy Duty Diesel Vehicle. *Environmental Science & Technology*, **41**, 6384–6389.

Schneider, I. L., Teixeira, E. C., Oliveira, L. F. S., & Wiegand, F. (2015). Atmospheric particle number concentration and size distribution in a traffic–impacted area. *Atmos. Pollut. Res.*, 6(5), 877-885.

Zhang, L., Gong, S., Padro, J., & Barrie, L. (2001). A size-segregated particle dry deposition scheme for an atmospheric aerosol module. *Atmos. Environ.*, 35(3), 549-560.

Zhong, J., Cai, X.-M. & Bloss, W. J. (2015). Modelling the dispersion and transport of reactive pollutants in a deep urban street canyon: Using large-eddy simulation. *Environ. Pollut.*, **200**, 42–52.

---

## Author Comment (AC2)

**Response to Referee 2**

We thank the referee for the interesting and helpful comments. Point-by-point responses are included below. For convenience, we first summarise our responses to the three major points and situate them with a broader scientific context.

1. *The time scale of mean circulation, i.e. residence time in the street canyon, of 380 s is very long in comparison to other published studies on street canyons.*

   Pollutant dispersion and aerosol dynamics are strongly influenced by the key dynamical timescales. As is generally the case in fluid dynamics, different timescales may be defined. Our choice of the mean canyon circulation timescale, $T_c$, is motivated by the finding that pollutant dispersion from relatively deep urban canopies (in the skimming flow regime) occurs on this timescale (Lo & Ngan 2017). Since $T_c$ approximates the e-folding timescale of the mean concentration and the mean age of air within a unit-aspect-ratio street canyon (Lo & Ngan 2016), it is a reasonable choice for this study of aerosol dynamics. One of our key findings is that coagulation within a representative urban canopy depends on the age of air or mean tracer age.

   The timescales quoted by the referee correspond to a different definition. The dilution timescale (Ketzel and Berkowicz 2004) characterizes the rate at which the volume of a plume changes (or equivalently the turbulent diffusion of a pollutant). Applying the version for an urban canopy (Nikolova et al. 2014), yields a much shorter timescale that agrees with previous studies.

2. *The formation of a stable vortex holds for neutral conditions, but it needs to be tested what consequences unstable conditions with thermal convection have on the concentration distribution in the street canyon.*

   It is certainly true that the occurrence of a central canyon vortex, which strongly influences the mean circulation and dynamical timescales, could be affected by unstable stratification. However, a canyon vortex does persist for a bottom-heated street canyon. This was first demonstrated in the two-dimensional Reynolds-Averaged Navier-Stokes calculations of Kim and Baik (2001). In a recent study, we have confirmed this using three-dimensional large-eddy simulation (Figure 1). The vortex persists over a wide range of bulk Richardson numbers, $-0.4 \leq Rb \leq 0.4$. From field measurements taken inside a real urban street canyon (Nakamura and Oke 1988), this range covers stable to moderately unstable conditions. Therefore, the assumption of neutral flow should generalize to representative urban conditions.

3. *Another aspect to consider is that when the wind is parallel with the street, the recirculating structure within the cavity disappears completely, and the concentration field becomes very different.*

   Yes, the flow structure and aerosol concentrations are very different for an external wind parallel to the canyon axis. However, this does not affect our

main finding, which is that the aerosol dynamics within the urban canopy depend on the ratio of the aerosol processes to the relevant dynamical timescales. As explained more fully in our response to Referee 1, the results described in the ms correspond to a regime in which the coagulation timescale is long relative to the dynamical timescale while the deposition timescale is of the same order of magnitude; therefore, both processes bear the imprint of the mean circulation, as may be seen in the relative difference fields. This claim has been verified by considering a parallel external wind. Although the structure of the passive scalar field changes from θ=0° to θ=90° (Figs. 2a,c), it does not follow that the qualitative effect of the aerosol dynamic processes also changes. At θ=90°, the number concentration field continues to reflect the structure of the mean circulation (Figs. 2b,d). This can be attributed to the ratios, $\tau_{\text{coag}}/T_{\text{c}}$ and $\tau_{\text{depo}}/T_{\text{c}}$, remaining qualitatively unchanged for θ=90°. While the absence of strong vertical motions increases the mean circulation timescale, the coagulation timescale continues to be much longer than the dynamical timescale and the deposition timescale continues to be slightly shorter (Table 1).

[Figure]

**Figure 1:** Spanwise-averaged vertical streamlines for a bottom-heated unit-aspect-ratio street canyon and an external wind perpendicular to the canyon axis (Wang et al. 2021). (a) Neutral flow (Rb=0); (b) Unstable flow (Rb=-0.39).

[Figure]

**Figure 2:** Comparison of results for different wind directions with respect to the axis of unit-aspect-ratio street canyon: (top) perpendicular ($\theta=0°$); (bottom) parallel ($\theta=90°$). (a,c) without microphysical processes; (b,d) with microphysical processes.

| $\theta$ | $\tau_{coag}/T_c$ | $\tau_{depo}/T_c$ |
|---|---|---|
| 0° | 3141 | 0.4 |
| 90° | 1529 | 0.5 |

**Table 1:** Comparison of timescales for $\theta=0°$ and 90°.

**General Comment**

**Comment**

This technical note deals with the spatial distribution of particles emitted from road traffic and from cooking sources in different vertical levels inside a street canyon. The PALM model coupled with the sectional aerosol dynamics model SALSA is used to determine the relevance of aerosol processes in a similar configuration as in Kurppa et al. (2019). From the analysis of time scales it is concluded that deposition mainly affects particles in the air close to the canyon surfaces, while the relevance of coagulation is related to the mean tracer age. Compared to Kurppa et al.

(2019) the novelty of the present work appears to be the consideration of particles emission spectra from kitchen exhaust ducts, which have a higher fraction of small particles than emission spectra from traffic.

The presentation of the Methodology part should be better organized, in particular with a separate section for the emission scenarios. The emission scenarios need to be in one place early in the method section because all the result sections are referring to the scenario abbreviations. The validation section is confusing.

**Response:** The emission scenarios are now defined in a separate subsection, Sec. 2.2.3, that precedes the results. We have confirmed that the scenario abbreviations are not referred to prior to this section.

Changes to the validation section are described under 'Specific Comments'.

**Comment**

My main concern is that the time scale of mean circulation, i.e. residence time in the street canyon, of 380 s is very long in comparison to other published studies on street canyons. For example, Nikolova et al. (2014) report CFD simulations of aerosol particles for a real street canyon in Antwerp having unit aspect ratio and a dilution time scale of 110 s for low wind. Ketzel and Berkowicz (2004) give dilution time scales within a range of 45−120 s. The long recirculation cycle does not seem realistic even at low winds, hence leading to an overestimation of the contribution of coagulation to the reduction of mean particle number concentrations compared to the case with no aerosol processes.

**Response:** The dilution timescale (Ketzel and Berkowicz 2004) differs from the mean circulation timescale calculated in the ms. In the version adopted by Nikolova et al. (2014), $T_{di}=10H/u_{roof}$, where $u_{roof}$ is the RMS streamwise velocity at the roof level. By contrast, we use $T_c = 2(\frac{W}{Urms} + \frac{U}{Wrms})$, where $u_{rms}$ and $w_{rms}$ are the RMS streamwise and vertical velocity over the entire canyon. Since it is defined using velocity data at the roof level, $T_{di}$ is a local timescale: the contributions of velocity fluctuations inside the canyon, and the influence of the topography, are essentially ignored. While $T_{di}$ may be more suitable for other applications, $T_c$ is a natural choice for a study focused on aerosol dynamics within the canyon. As noted above, there is a close connection between $Tc$ and the mean age of air. Nevertheless, we emphasize that the differences are a consequence of the definitions. Using the definition of Nikolova et al. (2014), we calculate $T_{di} \sim 60$ s, which falls within the range quoted by the referee. There is no reason to believe that the values of $T_c$, which are consistent with previous studies (e.g. Lo and Ngan 2017), are indicative of an unrealistic or unphysical flow regime.

**Comment**

The formation of a stable vortex holds for neutral conditions, but it needs to be tested what consequences unstable conditions with thermal convection have on the concentration distribution in the street canyon. Another aspect to consider is that when the wind is parallel with the street, the recirculating

structure within the cavity disappears completely, and the concentration field becomes very different. The authors should state such important limitations of the presented CFD simulations early in the text. Therefore, I suggest emphasizing that an idealized configuration of the street canyon was chosen, for the purpose of the study of particle emissions from different pollutant sources inside the street canyon.

**Response:** As explained above, the central canyon vortex persists under unstable stratification (Fig. 1). Furthermore, the effects of aerosol dynamic processes are qualitatively similar when the external wind is parallel rather than perpendicular to the canyon axis (Fig. 2). The relative importance of deposition or coagulation will change appreciably under two conditions. First, the residence time or mean age decreases dramatically so that $\tau_{coag}/T_c \lesssim 1$ or $\tau_{depo}/T_c \gtrsim 1$. Second, there is very intense local turbulence so that deposition or coagulation rates may be significantly larger within certain regions or structures. Given the persistence of the central canyon vortex for unstable stratification and the relatively insensitivity of the ratios to the wind direction (Table 1), we conclude that these conditions are not satisfied for coagulation, which is the dominant process for cooking emissions. With a much shorter dynamical timescale, so that $\tau_{depo}/T_c \gtrsim 1$, one expects the relative importance of coagulation to increase.

These issues are discussed in Section 6. See also our response to Referee 1.

**Specific comments**

1. Section 2.2.1: Provide more details on the configuration of the street canyon and mention which aspects are different to the street canyon simulated in the work of Kurppa et al. (2019).

**Response:** The street-canyon configuration is described in in Sec 2.2.1. The dimensions of the domain and topography are specified, as well as the grid spacing, boundary conditions, mean-flow forcing and stability. Details on the numerical schemes may be found in Sec. 2.1.1.

Differences with respect to the configuration of Kurppa et al. (2019) are now clearly stated (l.86):

> The configuration described above differs in several respects from K19. First, an idealised street canyon is used in place of realistic topography within a neighbourhood-scale domain. Second, there is uniform grid spacing rather than stretched grid. Third, the computational lid height of 5H is decreased from 13Havg, where Havg is the mean building height.

The most important difference is the substitution of a unit-aspect-ratio street canyon for realistic topography.

2. Section 2.2.2 has to be divided in a section on configuration of SALSA in this study and a section on emission scenarios.

**Response:** As mentioned above, a separate section on emission scenarios has been created.

3. More details on the coupling of SALSA with PALM need to be provided. For example, are the particle emissions entering into the SALSA model or first into the PALM model? Deposition of particles: only to street surface or also and wall surfaces; in which distance from the surface are particles affected by deposition? Condensation of which gases?

**Response:**

   i. Aerosol emissions are handled by SALSA rather than PALM. Hence a particle is subjected first to aerosol processes before being transported and advected by PALM. This is now explained on l. 100:

   > **Pollutants are emitted from uniform area sources. Since aerosol emissions are handled by SALSA rather than PALM, pollutants are subjected to aerosol dynamic processes before being transported and advected by PALM.**

   ii. Deposition occurs occur on all street and wall surfaces. Figure 13 shows that the deposition velocity is maximized in their immediate vicinity (i.e. within the first grid box). In fact, the deposition velocity vanishes away from the first grid box; therefore, only gravitational settling occurs at these points. This is now explained on l. 62:

   > **Briefly, deposition removes particles near surfaces; the deposition velocity is non-zero within the first grid box at a surface, e.g. from $z = 0$ to $z = \Delta z$ (cf. Fig. 13a); away from the surface, only gravitational settling occurs.**

   iii. Condensation of H2SO4, HNO3, NH3, semi-volatile (NVOCs) and non-volatile organics (SVOCs) is included. This is now mentioned explicitly on l.71:

   > **Gaseous components, namely H2SO4, HNO3, NH3, semivolatile (SVOCs) and non-volatile organics (NVOCs), may condense onto particles [...]**

4. Section 2.3: the presentation of the validation is unclear. Several references to figures are missing. Maybe first mention what kind of validations were performed, then describe each test in a separate paragraph.

**Response:**
Yes, the presentation could have been clearer. Following the referee's suggestion, several changes have been made:

   i. A new paragraph summarizing the different types of validation tests has been added to beginning of the section (l. 155).

**Several validation tests have been performed. First, the mean velocity statistics are validated against measurements of flow over parallel unit-aspect-ratio streets canyons (Brown et al., 2001). Second, passive scalar statistics are also validated (Pavageau and Schatzmann, 1999). Finally, the performance of the coupled PALM-SALSA model is compared to previous studies (Kumar et al., 2008; Kurppa et al., 2019).**

   ii.   Figure references have been added where necessary.
  iii.   The PALM and PALM-SALSA tests are described in separate paragraphs.

5.  P. 7 lines 148 - 153: explain the difference of the simplified computational domain. used in the validation and the computational domain in K19. Is the simplified computational domain intended to mimic the real street canyon in Cambridge? I think Figure 5 belongs to this validation, but it is not referenced here.

**Response:** Yes, the simplified computational domain is intended to mimic the real street canyon in Cambridge. The text has been modified to make this clearer (l. 149):

> **For simplicity, the computational domain is focused on this street canyon: no other buildings are included. In particular, a single street canyon of 167 m × 12 m × 12 m is centred inside a domain of 167 m × 60 m × 60 m.**

An explicit reference to Figure 5 is now included (l. 151): "Vertical profiles of the aerosol number concentration are compared in Fig. 5."

6.  P. 13 line 2: Figure 8 shows only boiling. Where is the figure panel for isolated kitchens, deep frying? Figure parts fig. 8a and 8d are not referenced in the text.

**Response:**   We have added results for isolated kitchens and deep-frying (I-D-z0) (now Fig. S-1 of the Supplementary Material). The corresponding values have been added to Table 3.

Table 3. As in Table 2, but for deep-frying and boiling emissions from isolated kitchens.

|          | NOAD         | AD           | difference |
|----------|--------------|--------------|------------|
| I-B-0.05 | $219.3 \pm 7.2$  | $200.0 \pm 5.5$  | -9%  |
| I-B-0.50 | $289.9 \pm 10.8$ | $276.2 \pm 8.7$  | -5%  |
| I-B-0.95 | $242.5 \pm 13.8$ | $231.4 \pm 11.8$ | -5%  |
| I-D-0.05 | $219.3 \pm 7.2$  | $181.3 \pm 3.0$  | -17% |
| I-D-0.50 | $289.9 \pm 10.8$ | $264.4 \pm 4.7$  | -9%  |
| I-D-0.95 | $242.5 \pm 13.8$ | $232.4 \pm 4.3$  | -4%  |

Figs. 8a and 8d are now referred to explicitly (l. 201).

> **Although trapping of particles within the vortex at the bottom leeward corner is less evident as the source height is increased from $z_0/H = 0.05$ to $z_0/H = 0.95$ (Figs. 8a-c),[...]. The vertical profiles (Fig. 8d) show [...]**

7. P. 14 lines 219 - 222: Time scale analysis for a street canyon in Cambridge by Kumar et al. (2008) reveals that dry deposition to road surface is much faster than deposition to wall surfaces. Can such a differentiation be made in this study as well?

**Response:** Yes, from the deposition velocity for NG-D (Figure 13), the deposition timescale may be estimated as $\Delta/v_d$, where $\Delta$ is the grid spacing. This shows that the deposition timescale for road and wall surfaces is ~65 s and ~110 s, respectively. In agreement with Kumar et al. 2008, deposition to the road surface is faster.

8. P.17 lines 254 – 256: The "plume-like structure" for case CO-D cannot be inferred from Figure 11. Should this refer back to Figure 7? It should be better indicated in the plot, how the plume like structure from column kitchen emissions develops.

**Response:** The "plume-like structure" refers to the tongue of low concentrations with RDcoag < 22% emanating from the column source. The colour bar and contour interval have been adjusted to show the plume-like structures in Figs. 11c,d more clearly. The text has also been modified to aid interpretation (l. 256):

> **however, the column source covers a larger area and a plume-like structure (i.e. the tongue of low RD values between the canyon centre and the windward wall) develops away from it.**

9. Section 3.4: it is not immediately clear where in the street canyon the aerosol number distribution were taken. If it is the canyon average distribution, then the standard deviations should be included in Figure 12. Where in the street canyon should measurements be done to be most sensitive to emissions of each of the different source types?

**Response:** The aerosol number distributions shown in Fig. 12 represent canyon averages. The caption has been updated to make this clear. Error bars have been added to depict the temporal standard error. The temporal standard error is chosen (rather than, for example, the spatial standard deviation) because it characterizes the error in the estimator rather than the spatial distribution. A new paragraph describing the statistical errors in the size distributions has been added (l. 270):

> **The uncertainty in the estimate of the time-averaged size distributions is indicated with the (temporal) standard error. Errors are much smaller for the deep-frying cases, NG-D and CO-D. A plausible explanation is that temporal intermittency is greater for cases in which deposition plays a more important role, namely TR, NG-B and CO-B, because deposition only occurs**

**near surfaces and is maximised inside the corner vortices. Coagulation, by contrast, occurs everywhere.**

The determination of appropriate measurement locations is an interesting question. We have not determined these locations precisely; however, approximate locations can be estimated from the relative difference fields of Sec. 3.3.

10. Section 5.1: the description of the background particles needs to be improved. How is the background aerosol mixed into the street canyon - is the spatial distribution the same as for the emissions or is the background entering from the boundaries? Did the simulations take into account heterogeneous coagulation between the emitted particles and the background particles, or were they assumed to be of the same population?

**Response:** The background concentrations for the entire domain are fixed; hence, there is no spatial distribution. Yes, heterogeneous coagulation between the emitted particles and the background particles is included: to a large extent, the results described in Sec. 5.1 concern implications of this phenomenon.

To avoid confusion, the uniform nature of the background concentrations is now noted (l. 331):
> **Spatially uniform, constant background concentrations are prescribed over the entire computational domain.**

The inclusion of heterogeneous coagulation is also noted (l. 332, l. 336):
> **Note that the background is allowed to interact with the emissions through heterogeneous coagulation.[...] On account of aerosol processes involving the background only or the background and emission [...].**

We believe that the description of the background particles should now be clear.

**Technical Corrections**

P. 7 lines 139: correct "ine Appendix B".

Fixed.

P. 14 line 229: deep frying?

Fixed.

P. 26 line 431: delete "happens".

Fixed.

**References**

Ketzel, M., and R. Berkowicz, 2004: Modelling the fate of ultrafine particles from exhaust pipe to rural background: an analysis of time scales for dilution, coagulation and deposition. *Atmos. Environ.*, **38**, 2639–2652.

Kim, J.J. and Baik, J.J., 2001. Urban street-canyon flows with bottom heating. *Atmos. Environ.*, **35(20)**, 3395-3404.

Lo, K. W., and Ngan, K. 2015: Characterising the pollutant ventilation characteristics of street canyons using the tracer age and age spectrum. *Atmos. Environ.*, **122**, 611–621, https://doi.org/10.1016/j.atmosenv.2015.10.023.

Lo, K. W., and   Ngan, K. 2017: Characterising urban ventilation and exposure using Lagrangian particles. *J. Appl. Meteorol. Climatol.*, **56**, 1177–1194, https://doi.org/10.1175/JAMC-D-16-0168.1.

Nakamura, Y. and Oke, T.R., 1988. Wind, temperature and stability conditions in an east-west oriented urban canyon. *Atmos. Environ.* (1967), **22(12)**, 2691-2700.

Nikolova, I., S. Janssen, P. Vos, K. Vrancken, V. Mishra, and P. Berghmans, 2011: Dispersion modelling of traffic induced ultrafine particles in a street canyon in Antwerp, Belgium and comparison with observations. *Sci. Total Environ.*, **412**, 336–343.

Nikolova, I., Janssen, S., Vos, P.V. and Berghmans, P., 2014. Modelling the mixing of size resolved traffic induced and background ultrafine particles from an urban street canyon to adjacent backyards. *Aerosol Air Qual. Res.*, **14(1)**, 145-155.

Wang, H., Furtak-Cole, E. and Ngan, K. 2021: Predicting mean wind profiles inside realistic urban canopies, J. Wind Eng. & Ind. Aerodyn. submitted.

---

## Author Response (AR2)

**Response to Referee 1**

We thank the referee for the helpful comment.

> Yes, OK but then how do you estimate/calculate the equilibrium saturation ratio and saturation vapour mole concentrations in the model? What saturation vapour mole concentrations did the different vapors have? I think this information I miss in the manuscript and appendix A.

**Response:** We calculated the equilibrium saturation ratio and saturation vapour mole concentration following Jacobson et al. 2005.

Briefly, the equilibrium saturation ratio is calculated as $exp(4\sigma m/(DR^*T\rho))$, where $\sigma$ is the average particle surface tension, $m$ is the average particle molecular weight, $D$ is the particle diameter, $R^*$ is the universal gas constant, $T$ is the temperature, and $\rho$ is the average particle density. The uncorrected saturation vapour mole concentration is calculated as $p/(R^*T)$, where $p$ is the saturation vapor pressure. The saturation vapour mole concentration is given by the product of the equilibrium saturation ratio and the uncorrected saturation vapour mole concentration. Since the equilibrium saturation ratio and saturation vapour mole concentration are related to particle features, their values for different vapours also depend on the size bin described in the configuration part. Details on the gaseous components can be found in Section 2.1.2.

The relevant equations have been added to Appendix A.

**Response to Referee 2**

We thank the referee for bringing these technical points to our attention.

Page 8, line 160: please add a reference to Figure 4.

**Response:** An explicit reference to Figure 4 is now included (l. 160): "as shown in Fig 4, vertical profiles of the mean streamwise velocity u show good agreement ..."

Page 8, line 164: configuration details in section 2.2.

**Response:** Fixed.

Page 27, line 451: it would be good to include a sentence about the much lower relevance of coagulation for the traffic emissions, compared to the cooking emissions.

**Response:** Thank you for the suggestion. The text (p.27, l451) has been modified as follows:

[revised manuscript text omitted]

**Table S-1.** Aerosol timescales for $0°$ and $90°$.

[Figure]

**Figure S-2.** As in Fig. 9, but for Case NG-B.

(a)

(b)

[Figure]

**Figure S-3.** As in Fig. 9, but for case CO-B.

[Figure]

**Figure S-4.** Vertical profiles of the mean number concentration for emission scenario NG-B and all aerosol processes for the default emission spectrum (ALL); displacement to large scales by a factor of 2 (ALL-LD); and displacement to small scales by a factor of 0.5 (ALL-SD).

[Figure]

**Figure S-5.** Relative difference fields for NG-B: (a) displacement to small scales, SD; (b) default emission spectrum; (c) displacement to large scales, LD.